# A bilateral interfacial passivation strategy promoting efficiency and stability of perovskite quantum dot light-emitting diodes

Leimeng Xu[1,2,3], Jianhai Li[1,2,3], Bo Cai [1,2,3], Jizhong Song [1,2✉], Fengjuan Zhang[1,2], Tao Fang[1,2] & Haibo Zeng [1,2✉]

Perovskite quantum-dot-based light-emitting diodes (QLEDs) possess the features of wide gamut and real color expression, which have been considered as candidates for high-quality lightings and displays. However, massive defects are prone to be reproduced during the quantum dot (QD) film assembly, which would sorely affect carrier injection, transportation and recombination, and finally degrade QLED performances. Here, we propose a bilateral passivation strategy through passivating both top and bottom interfaces of QD film with organic molecules, which has drastically enhanced the efficiency and stability of perovskite QLEDs. Various molecules were applied, and comparison experiments were conducted to verify the necessity of passivation on both interfaces. Eventually, the passivated device achieves a maximum external quantum efficiency (EQE) of 18.7% and current efficiency of 75 cd A$^{-1}$. Moreover, the operational lifetime of QLEDs is enhanced by 20-fold, reaching 15.8 h. These findings highlight the importance of interface passivation for efficient and stable QD-based optoelectronic devices.

[1] School of Materials Science and Engineering, Nanjing University of Science and Technology, 210094 Nanjing, China. [2] MIIT Key Laboratory of Advanced Display Materials and Devices, Institute of Optoelectronics & Nanomaterials, 210094 Nanjing, China. [3]These authors contributed equally: Leimeng Xu, Jianhai Li, Bo Cai. ✉email: songjizhong@njust.edu.cn; zeng.haibo@njust.edu.cn

Lead halide perovskites possess photoluminescence quantum yields (PLQYs) as high as 100% and narrow full width at half maximum (FWHM) of around 20 nm, which makes them high-profile candidates for high-quality lightings and displays[1–6]. Up to now, major breakthroughs have been made in perovskite-based light light-emitting diodes (LEDs), of which film-based LEDs have achieved 20.3% and 21.6% of EQE at green and infrared region, respectively[7,8]. As another vital part of perovskite LEDs, QD-based LED (QLED) has obtained extensive attention for their more flexible solution processing characteristics and better mass production potential. Since the first device reported in 2015[4,9–11], perovskite QLEDs have also made great breakthroughs during the past years. For example, the red perovskite QLEDs exhibited an EQE of 21.3% through an anion-exchange[12], which has surpassed the film-based LED in red region[13]. Howbeit, the maximum EQE of blue and green QLEDs is 2.8%[14] and 16.48%[15], respectively, which is lower than that of film-based perovskite LEDs[16,17]. Consequently, it is highly desired to explore a feasible and effective strategy to preparing the highly efficient perovskite QLEDs.

In general, highly efficient exciton recombination in QD films are significantly critical for high-performance QLEDs. Highly luminescent perovskite QD films are determined by the quality of QD materials and film constructions. As the central role, a lot of work has been devoted to optimize the quality of colloidal QDs, including component regulation, surface engineering and other process optimization, which are effective to enhance the radiative recombination of perovskite QD films. For example, the PL properties of $CsPbX_3$ QDs can be improved by alloying A-site cation of FA and MA[18,19], or doping the B-site metal cation of $Sn^{2+}$, $Mn^{2+}$, $Ce^{3+}$[20–22]. In addition, massive hanging bonds or defects[23,24] on QDs would reduce exciton recombination efficiency. In this regard, some surface passivation routes were applied, such as introducing the organic ligand of didodecyl dimethyl ammonium bromide (DDAB)[25,26], or passivating the QDs with inorganic ligands[27,28]. The above-mentioned methods all focus on the improvement of colloidal QDs, ignoring the damage that the film-forming process may bring to QDs to a certain extent. Therefore, it is also crucial to improve device performance from the perspective of QD films.

As is known that a large part of fluorescence is lost when the colloidal QDs transform into the QD solids, this is because massive defects would be introduced inevitably during the film-forming process[29–31]. These defects are prone to regenerate during the device construction process on account of the highly sensitive surface and complex interface environment, and hence lead to the formation of non-radiative recombination centers. In addition, these defects located on the interface between QD layer and carrier-transporting layers would sorely affect the injection and transportation of carriers, and degrade the device efficiencies[32–34]. To solve the problem, a lot of film-treatment methods have been applied to passivate the QD film, such as proper oxygen treatment, solvent treatment, or coating organic molecules on the interface[35–38]. Interface molecular passivation has been widely used in perovskite-based device, which could not only improve the effective radiation recombination, but also enhance the stability[6,8,39]. However, most of work only focused on the top surface of the perovskite film, few reports noticed the importance of double-sided passivation. However, it is well known that the perovskite layer is at the center of the sandwich structure in practical optoelectronic devices, both the top and bottom surface of the QD film may face the interface problems that defects and other deposited materials can affect the carrier behavior inside the film. Thus, the interface treatment on both sides of perovskite QD film may provide a good way for the depressed device efficiency and stability.

In this work, we present a bilateral passivation strategy to reduce the interfacial defects of perovskite QD film, through evaporating a layer of organic molecules between QD films and carrier transport layer (CTL). The phosphine oxide molecule, diphenylphosphine oxide-4-(triphenylsilyl)phenyl (TSPO1), was used as the typical passivation molecule. The density functional theory (DFT) calculations were used to reveal the decreased defect traps and non-radiative recombination. The decreased defects were further verified by transient TA spectra analysis and space charge-limited-current (SCLC) method, and the improved exciton recombination efficiency is reflected in the increased PLQY of QD film (increase from 43 to 79%) and increased electro-optic conversion efficiency (the current efficiency of QLEDs increase from 20 to 75 cd $A^{-1}$, and the maximum EQE from 7.7 to 18.7%). In addition, the comparison experiments of unilateral and bilateral passivation were conducted to demonstrate the necessity of passivating both interfaces. Besides TSPO1, a series of other organic molecules used in this system also achieved impressive results, which showed the universality of this bilateral passivation method. Moreover, profiting from the strong interaction with perovskite and blocking between perovskite and CTL, bilateral-passivated molecules endow the films and LEDs with enhanced stability. For example, a 20-fold enhancement in the $T_{50}$ operational lifetime (from 0.8 h to 15.8 h) was observed. Our study demonstrates that defects on the interface between the QD films and charge transporting layers are detrimental for devices, which can be hopefully suppressed by bilateral passivation. The findings highlight the importance of passivation on both interfaces of QD films for constructing high-performance perovsktie QLEDs as well as other QD-based optoelectronic devices, including solar cells, and phodetectors.

## Results

**Bilateral passivation strategy and theoretical model**. The high density of dangling bonds and uncoordinated atoms (e.g. Pb and/or halide vacancy) caused by solvent evaporation and lost surface ligands is responsible for the traps and non-radiative recombination, which degrade the PL emission and device performances[40,41]. In this regard, we propose a bilateral passivation strategy through evaporating organic ligands on both top- and bottom-side of perovskite QD film, the theoretical model passivated with TSPO1 as typical ligand was shown in Fig. 1a. The interaction between uncoordinated Pb and P = O from TSPO1 was deemed to be the key factor that passivate the defects of QD films, which would be further discussed later. In the device, those defects mentioned above would capture electrons and holes, thus resulting in low device efficiency (Fig. 1b). Meanwhile, these defects could provide ion migration channel that might accelerate the degeneration of device, causing the poor stability[42]. Based on the bilateral-passivated QD film, we further designed the optimized device with TSPO1 on both top- and bottom-side of QD films which was expected to passivate the defects, decrease non-radiative recombination, and thus improve the device performances (Fig. 1c).

The theoretical calculation based on aforementioned model in Fig. 1a was applied to verify the conjecture. The forming energy between Pb and O from TSPO1 was −1.1 eV calculated by density functional theory (DFT), which showed that the interaction between surface Pb and O = P could be easily formed. Furthermore, the calculated density of states (DOS) was also applied to assess the passivation action of TSPO1. We performed DFT calculations assuming a $PbBr_2$-rich $CsPbBr_3$ surface, in which non-coordinating Pb atom located outside surface. The DOS curves of the TSPO1-passivated and pristine structures are reversed to provide an intuitive comparison plotted in Fig. 1d and

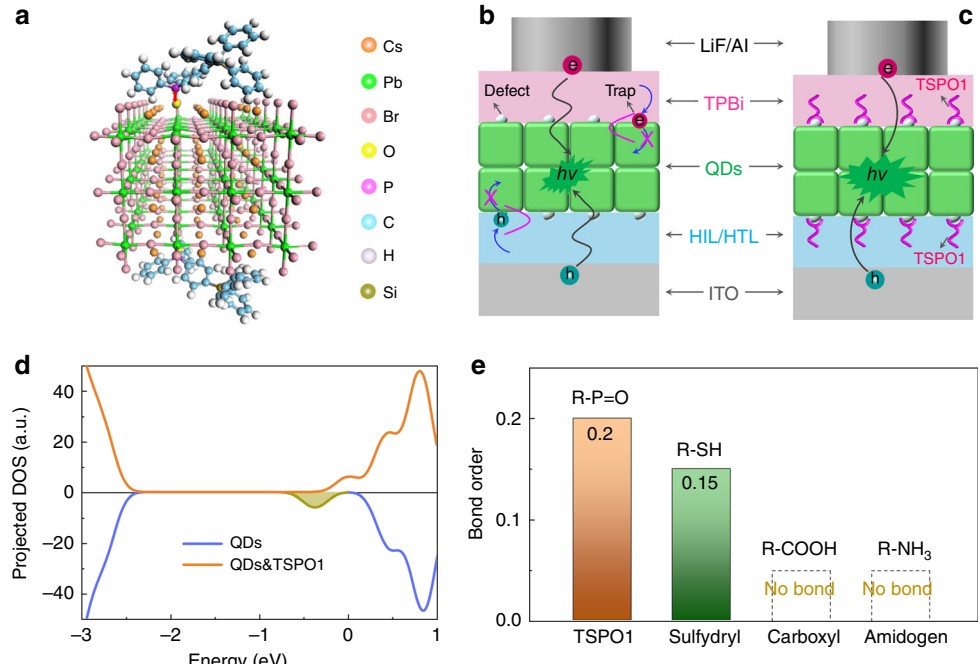

**Fig. 1 The bilateral passivation strategy for efficient and stable QLEDs. a** Diagram of the interaction between TSPO1 and perovskites. The red double-headed arrow show that the TSPO1 group has interaction with the uncoordinated Pb atom. The structure of QLED based on QD films passivated without (**b**) and with passivation (**c**), schematic illustration that TSPO1 could passivate defects on the surface of QD films, the defects may trap carriers (e.g. holes, electrons), decrease exciton recombination, and hence degrade the device performances. **d** DOS of the perovskite surface passivated with and without passivation. **e** Calculated bond order of surface Pb atom with TSPO1, sulfydryl, carboxyl, and amidogen ligands respectively. Source data are provided as a Source Data file.

Supplementary Fig. 1. The DOS of the unpassivated surface showed significant trap states on the band edge, which is due to the non-coordinating Pb atom. While, the trap states in QD films after passivating by TSPO1 was greatly weaken, which indicated the TSPO1 could effectively passivate defects, eliminate the trap states, and prevent the trapping of carriers.

In addition to passivating defects, the $P = O$ has strong interaction with surface Pb atom, which could prevent the loss of ligands from electric field. The weak adhesion of surface ligands on perovskite QD is responsible for defect regeneration[43]. We compared the bond order between surface Pb atom and several functional group of common organic ligands (TSPO1, sulfydryl, carboxyl and amidogen ligands), as shown in Fig. 1e. It could be seen that carboxyl and amidogen ligands (such as oleic acid and oleylamine) cannot bond with Pb atom, which is responsible for the poor stability and attenuating luminescence. However, phosphorus oxygen groups exhibited stronger interaction with Pb (bond order is 0.2) compared to other groups, which could suppress the regeneration of defects. The regenerative defects might act as the ion migration channel that was detrimental to stability. Meanwhile, the tough passivation layer could also block the ion migration and damage to perovskite from the transport materials at the interface, leading to better device stability.

**Characterization of perovskite QD films**. We investigated perovskite films based on CsPbBr$_3$ QDs synthesized by the typical hot-injection method[3,4]. The as-synthesized QDs had a cubic and uniform morphology with the size of 8 nm and the QD ink exhibit excellent PL properties with a PLQY of 85 ± 3% and a FWHM of 20 nm (Supplementary Fig. 2). Nevertheless, the light emission of the film exhibited a sharp decline due to the formation of non-radiative recombination centers during the film-forming process. The QD films are shown in Fig. 2a, and the schematic diagrams

present the passivating location of the TSPO1 molecules. The photogragh of QD film under UV lamp exhibits that the naked CsPbBr$_3$ QD film shows a dim glow compared to colloidal QDs. While, the emission is significantly enhanced after passivating TSPO1 at the interface of QD film, the up-side passivation shows better improvement than bottom passivation because the exposed upper surface faces severer challenge. Eventually, the bilateral-passivated film exhibits the brightest glow. It means that the passivation on both up- and bottom-side of QD film is necessary, traps are effectively passivated by TSPO1. The interaction between TSPO1 and perovskite was confirmed by Fourier transform infrared (FTIR) spectroscopy (Fig. 2b), in which the characteristic peak at 1188 cm$^{-1}$ was observed for TSPO1, corresponding to the stretching vibrations of $P = O$ bond. It can be found that the $P = O$ peak drifts to about 1184 cm$^{-1}$ in the QD films, which indicates that the bonding between TSPO1 and perovskite QD can be formed[6,44]. In addition, X-ray photoelectron spectroscopy (XPS) results showed that Pb 4 f peak shifts towards higher binding energy (BE) of about 0.2 eV for TSPO1-passivated QD films in reference to unpassivated one (Fig. 2c). It also unveiled the chemical bonding between $P = O$ group and Pb atom, which is due to the coordination between electronegative O$^-$ and uncoordinated Pb$^{2+}$, leading to higher binding energy for Pb 4 f. This showed that Pb atom on QD films indeed had interaction with O of TSPO1, which was consistent with the calculated results.

The steady-state PL spectra of above QD films were presented in Fig. 2d, which showed that PL intensity of the QD films were greatly enhanced with PL peaks remaining unchanged after TSPO1 passivation. It means that the TSPO1 effectively improve the radiative recombination without altering the structure of perovskite QDs. The absolute PLQY of QD films under different passivation strategy were tested to further evidence the increased exciton recombination efficiency, as shown in Fig. 2e. It could be seen that PLQY of naked QD film exhibited a sharp decline to

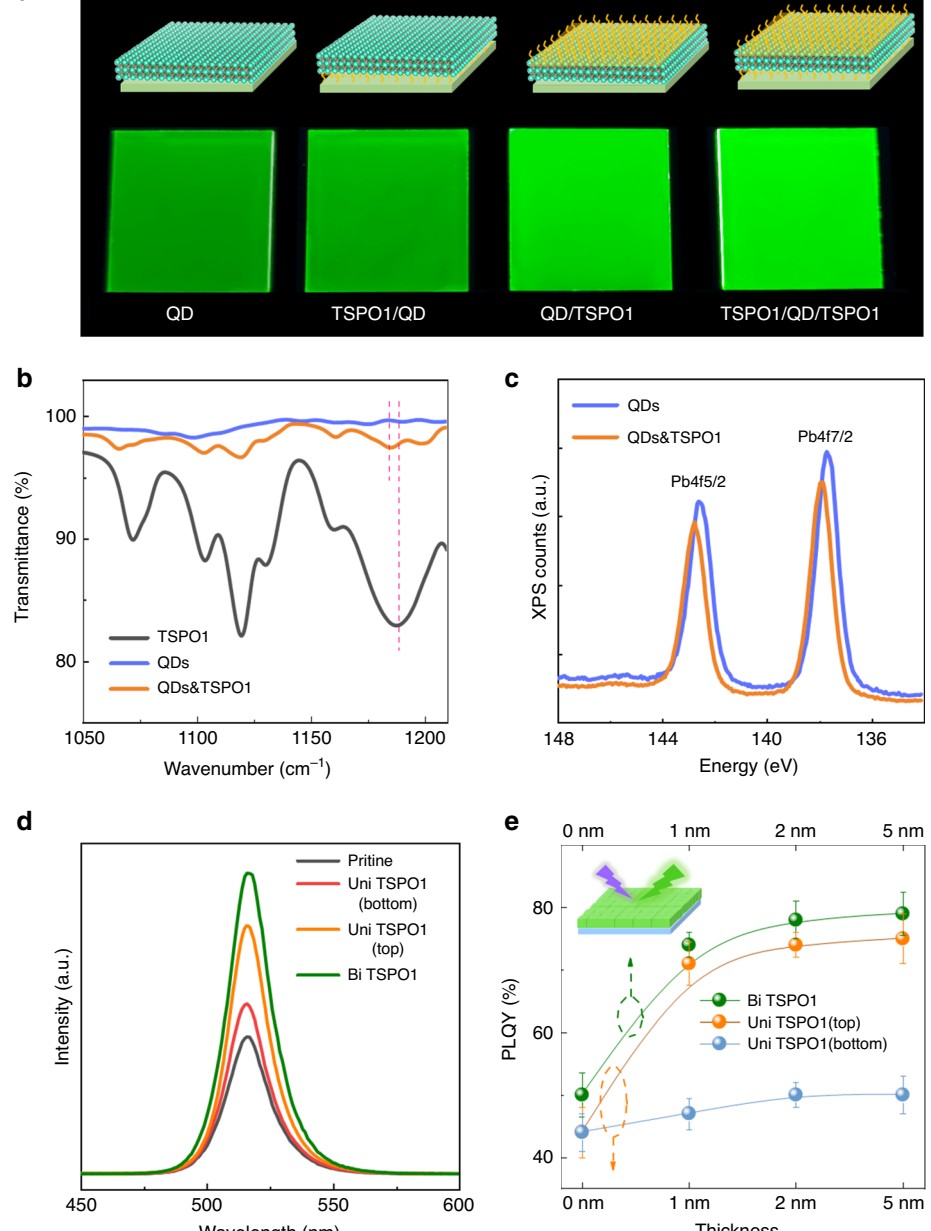

**Fig. 2 PL property comparison of QD films with different passivation states. a** Schematic diagram and photograph of QD films without passivation and with TSPO1 on the bottom side, on the top side, on both sides of QD film under UV light. **b** Fourier transform infrared (FTIR) spectroscopy measurement for TSPO1, QD and TSPO1-QD films prepared on silicon wafers. **c** Pb 4 f core level XPS spectra of primal and TSPO1-passivated CsPbBr₃ QD films. **d** PL spectra and (**e**) PLQY of QD films without and with TSPO1 on the bottom side, on the top side, on both sides of QD film. Error bars represent standard deviation of experimental data acquired from three times. Source data are provided as a Source Data file.

43 ± 4% versus colloidal QDs. The low PLQY could be greatly improved with increasing TSPO1 passivation thickness. However, the optimization effect was saturated when the TSPO1 thickness reached a certain value because the defects were passivated adequately at this level. Compared to passivation on bottom, top-side passivaion exhibited better optimization due to the exposed upper surface. While, unilateral passivation only was not as effective as the bilateral passivation. The maximum PLQY of QD film could be increased to 79 ± 3% under bilateral passivation, which indicated more efficient electron−hole recombination in the passivated QD films. From above results, although single-side interface passivation can also improve the performance of QD film, treatment on both sides is more effective.

**Reduced non-radiative recombination in bilateral-passivated perovskite QD film.** Ultrafast exciton dynamics analysis was carried out to make a further insight into the promotion effect of TSPO1 passivation for PL properties. Both QD films showed photo-bleaching peaks at around 515 nm (Supplementary Fig. 3a and 3b) corresponding to their energy band structure, which are consistent with the steady-state PL peak. From the entire bleaching color mapping, no extra bleaching signal appears after passivated by TSPO1, indicating no associated energy transfer process occur between TSPO1 and QDs. In order to parse these blending spectral profiles, the characteristic decay-associated spectra (DAS) of QD film was obtained through a global fitting analysis (Fig. 3a, b). The extracted time constants were as follows:

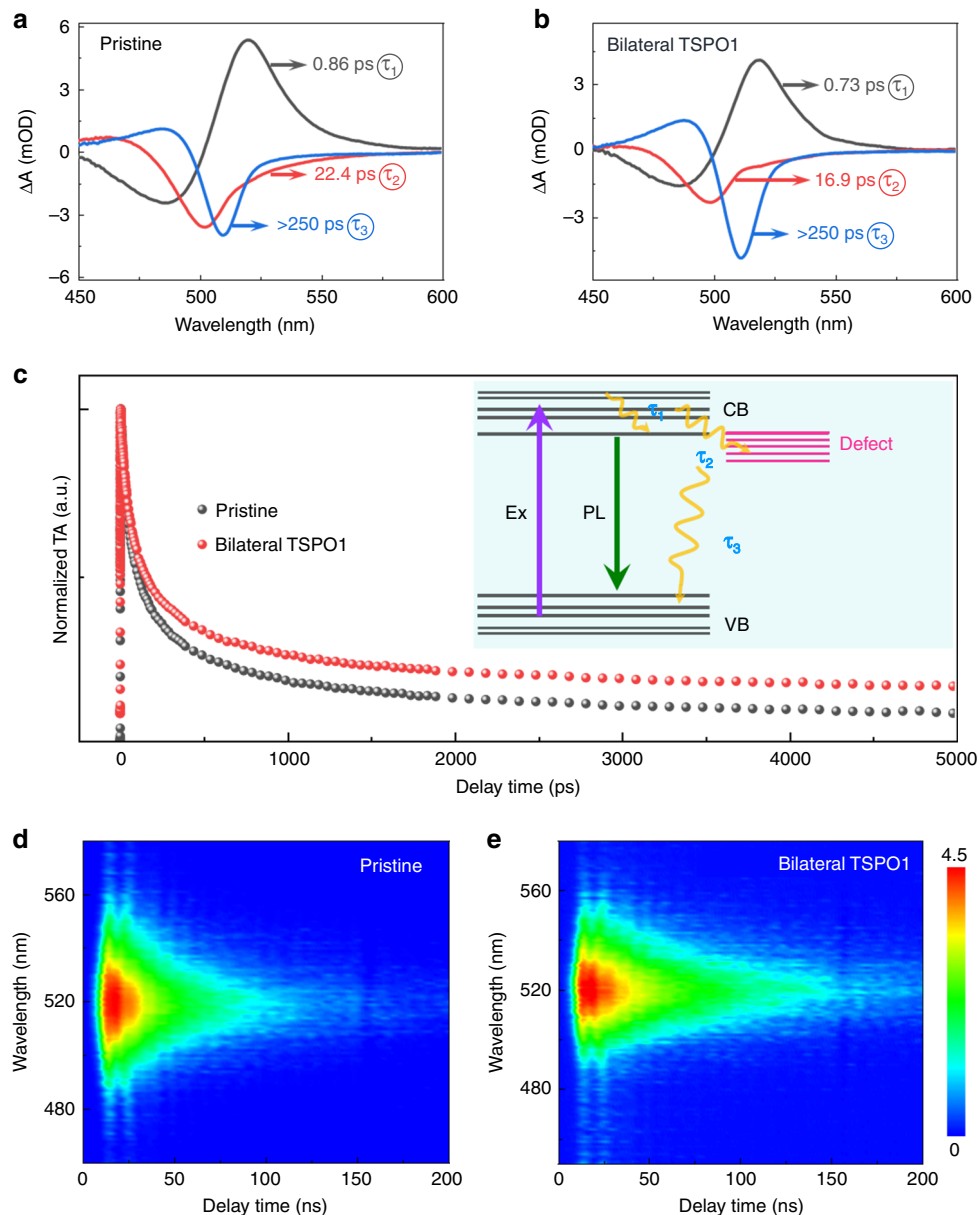

**Fig. 3 Exciton dynamics comparison of QD films with different passivation states. a, b** Decay-associated spectra for pristine and TSPO1-passivated QD films. The processes of intraband hot-exciton relaxation ($\tau_1$), exciton trapping to the band-edge trap states ($\tau_2$), and exciton recombination ($\tau_3$) was extracted to study the exciton dynamics. **c** Comparison of transient TA spectra with an excitation fluence of 5 μJ cm$^{-2}$. Inset: Schematic illustration of the photoinduced relaxation processes involved in the QD films. Two-dimensional contour image of time-resolved PL delay for (**d**) pristine and (**e**) bilateral TSPO1-passivated QD films. Source data are provided as a Source Data file.

$\tau_1 = 0.86 \pm 0.01$ ps, $\tau_2 = 22.41 \pm 0.11$ ps, and $\tau_3 > 250$ ps for the pristine QD films, while $\tau_1 = 0.73 \pm 0.02$ ps, $\tau_2 = 16.91 \pm 0.12$ ps, and $\tau_3 > 250$ ps for the passivated sample. The above decay parameters mainly express the following three processes of excitons:[20,45] hot-exciton relaxation process in the band ($\tau_1$), exciton trapping to the trap states at band edge ($\tau_2$), exciton recombination ($\tau_3$) as presented in Fig. 3c inset. Apparently, the first two processes of the passivated sample were accelerated compared to the unpassivated one, which demonstrated that TSPO1 facilitated the state coupling related to the relaxation processes. The TSPO1-passivated QD films with a shorter lifetime of $\tau_1$ and $\tau_2$ indirectly indicated the $\tau_3$ would be longer. The comparison of decay time $\tau_3$ indicated that the TSPO1-passivated QD films exhibited a slower kinetic recombination delay (Fig. 3c), which reflected a lower density of surface defect trap states in the

films. Thus, the surface defect related non-radiative recombination was suppressed by TSPO1.

Typically, the trap states on the interface of QD films would increase the non-radiative recombination of excitons, leading to the shorten lifetime. Therefore, passivating the interface of QD films would make a profitable impact on the carrier lifetime, thus conduce to the prolonged exciton lifetime. In order to compare the difference, time-resolved PL mapping of QD films was conducted as shown in Fig. 3d, e. The conspicuous improvement in the PL decay lifetime absolutely indicated surface defect passivated by TSPO1. Resultantly, the TSPO1-passivated films showed an average lifetime of approximately 13.9 ns longer than that of approximately 6.7 ns for primal QD films (Supplementary Fig. 4). These findings imply that TSPO1 effectually reduce the non-radiative recombination centers on the QD films.

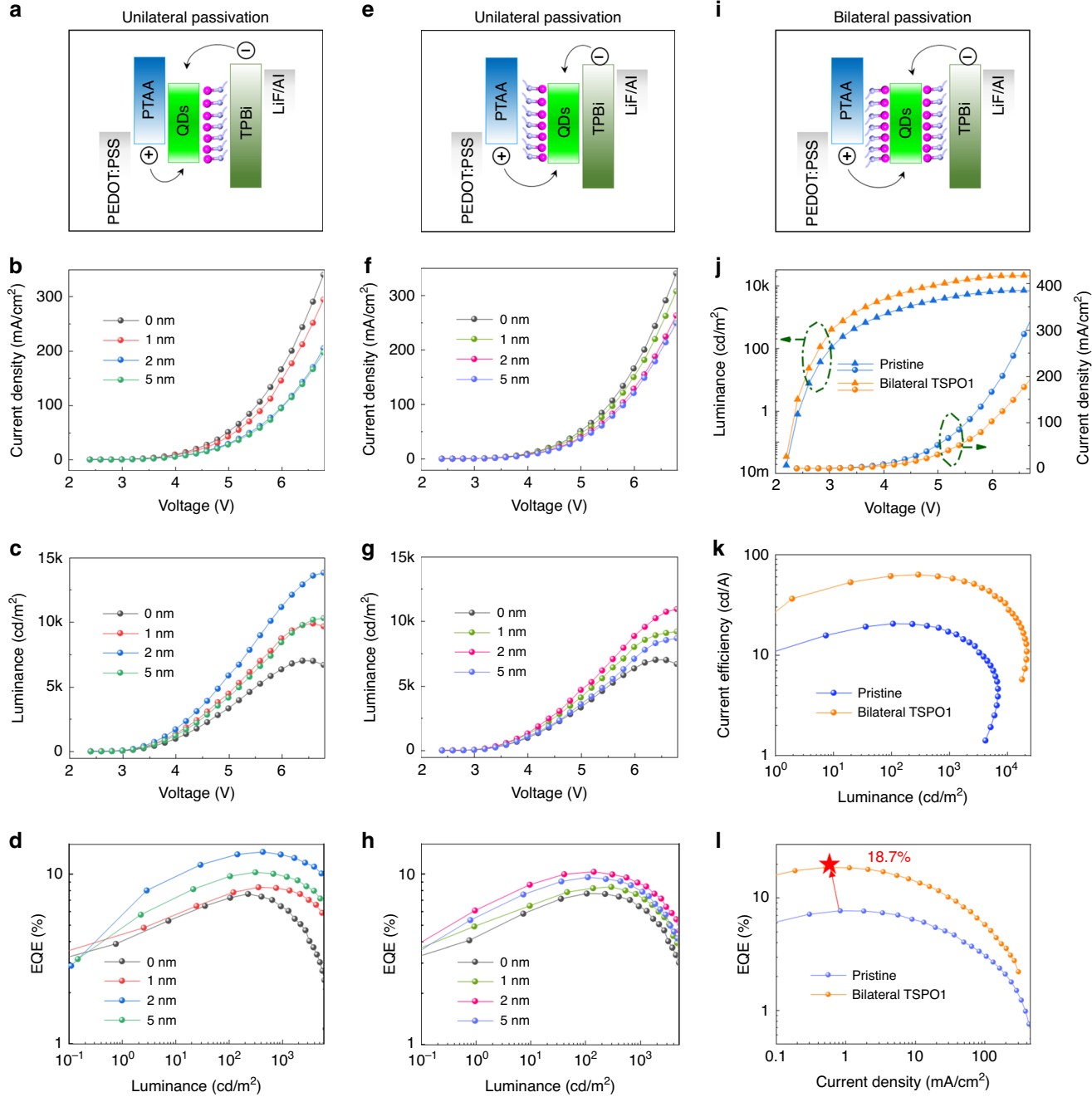

**Fig. 4 EL performance comparison of QLED with different passivation states. a** Energy level diagram, (**b**) current density vs. driving voltage, (**c**) luminance vs. driving voltage, and (**d**) EQE vs. luminance of device with different thickness of TSPO1 on the top side of QD layer. **e** Energy level diagram, (**f**) current density vs. driving voltage, (**g**) luminance vs. driving voltage, and (**h**) EQE vs. luminance of device with different thickness of TSPO1 on the bottom side of QD layer. **i** Energy band diagram of the bilateral-passivated device. The comparison of (**j**) current density and luminance, (**k**) current efficiency, and (**l**) EQE of original device and bilateral-passivated device. Source data are provided as a Source Data file.

**Performance improvement in bilateral-passivated perovskite QLED.** The QLED device structure with multilayer of indium tin oxide (ITO)/poly(3,4-ethylenedioxythiophene):poly-(styrenesulfonate) (PEDOT:PSS)/poly[*bis*(4-phenyl)(2,4,6-trimethylphenyl) amine] (PTAA)/QDs/1,3,5-*Tris*(1phen-y-1H-benzimidazol-2-yl) benzene (TPBi)/LiF/Al is used to evaluate the electroluminescence performance (Fig. 4). In order to demonstrate the necessity of bilateral passivation, three passivated structure, QDs/TSPO1, TSPO1/QDs, TSPO1/QDs/TSPO1, were designed in QLEDs. Figure 4a–d presented the carrier transportation and recombination process of unilateral-passivation only device

(QDs/TSPO1). We observed the increasing TSPO1 thickness partly decreased the electrical properties of the device possibly due to the relative poor carrier mobility, compared with the TPBi (Fig. 4b and Supplementary Fig. 5). However, the luminance was obviously enhanced under TSPO1 passivation, which reflected the higher exciton recombination efficiency to some extent (Fig. 4c). In particular, the device with 2 nm TSPO1 exhibits the highest luminance, which reaches a maximal value greater than 14000 cd m$^{-2}$ at 6.8 V. The device after passivation exhibited higher luminance and lower current density under the same voltage driven, which indicated the device had a higher electro-optic

conversion efficiency. For example, the current efficiency of the device with 2 nm TSPO1 was 63.1 cd A$^{-1}$ (Supplementary Fig. 6), higher than that of the control device. Accordingly, the passivated device exhibited a higher EQE with a peak value of 13.5% in Fig. 4d, which was higher than the un-passivated one (EQE of 7.7%). It could be seen that TSPO1 passivation could effectively improve the radiative recombination and enhance the performance of perovskite QLEDs.

Correspondingly, the unilateral passivation only on the other interface of QD film (TSPO1/QDs) was also constructed as shown in Fig. 4e. The current density decreased with increasing thickness of passivation layer, which exhibited the same trend as the foregoing results. While the brightness of the device was enhanced profiting from the effective passivation by TSPO1, the maximum luminance reach 11000 cd m$^{-2}$ at 6.8 V when the thickness of TSPO1 was 2 nm. The bottom-passivated device also obtained improved electro-optic conversion efficiency that was reflected on lifted current efficiency and EQE in Supplementary Fig. 7 and Fig. 4h. Under the 2 nm TSPO1 between QDs and PTAA, the current efficiency increased to 43 cd A$^{-1}$, and EQE increased to 10.2%. Compared to TSPO1 passivation between QD and TPBi, the PTAA/TSPO1/QD model did not achieve the same significant optimization effect, because TSPO1 was reported to be a partial electron transport material[46] that made it perform better at the interface between QDs and ETL. From the above results, either the upper-side or the bottom-side passivation by TSPO1 can successfully improve the device efficiency, which means that passivation on both upper and lower interfaces of QD film is necessary.

Indeed, unilateral passivation has achieved good results, however, if we passivate the top and bottom interfaces of the QD film with TSPO1 simultaneously, can we improve the performance of the device by another level? Therefore, we further constructed the bilateral passivation devices with 2 nm TSPO1 on both upper and lower sides of perovskite QD film as shown in Fig. 4i. As expected, the bilateral TSPO1 passivation made remarkable improvement in device performance (Fig. 4j–l). Figure 4j presented the current density and luminance vs. voltage curves, bilateral TSPO1 further reduced the current density for the lower carrier transport ability of TSPO1. Nevertheless, the exciton recombination efficiency could be greatly improved through bilateral passivation. The maximal luminance was enhanced to 21,000 cd cm$^{-2}$, which is much higher than unilateral passivated devices. Meanwhile, the current efficiency was increased to 75 cd A$^{-1}$ (Fig. 4k), and the maximal EQE reached 18.7%, which achieved an improvement of 140% compared to unpassivated device (Fig. 4l). The EL spectra did not shift after passivation, the optimized devices and controlled one both exhibit the EL peak centered at 516 nm with a FWHM of 20 nm (Supplementary Fig. 8). It indicated that the passivation layer would not affect the crystal structure of perovskite. Furthermore, no spectrum drift was observed under the driving of different voltages (Supplementary Fig. 9). It can be seen that interface passivation is an available strategy to improve the device performance, and compared to unilateral passivation, bilateral passivation provides a better option for highly efficient LEDs.

**Performance of bilateral-passivated perovskite QLEDs with various organic molecules**. Here, we were trying to offer a universal bilateral passivation strategy for perovskite LEDs, thus, besides TSPO1, a series of organic molecules with various functional groups, including DPEPO, TPPO (~P = O), DMAC-DPS (~S = O), nitrosobenzene (~N = O) and benzophenone (~C = O), were also applied in the bilateral-passivated device structure. Figure 5a exhibited the schematic diagram of the bilateral-

passivated device structure and its corresponding TEM image of the sectional view. The schematic structures of these organic molecules are listed in Fig. 5b and Supplementary Fig. 10. In addition to N = O would seriously damage the perovskite QDs and QD film, other molecules could improve the QLED performances to varying degrees (Fig. 5c, d and Supplementary Fig. 11), of which P = O and S = O performed better. Compared to controlled device, the bilateral-passivated QLEDs with P = O and S = O based molecules showed higher brightness under same current density (Fig. 5c), which demonstrated increased radiative recombination. Meanwhile, the improved conversion efficiency was revealed by the average peak current efficiency and EQE (Fig. 5d, e) from 20 devices. The device performance parameters through bilateral passivation strategy with these organic molecules were summarized in Table 1. The current efficiency of passivated devices was over 60 cd A$^{-1}$, and the average EQE of devices passivated by TSPO1, DPEPO and TPPO was over 15%. These results indicate bilateral passivation strategy are generalized and feasible for passivating the defects on the surface of QD films, and increasing the exciton recombination.

**Improved stability of bilateral-passivated QLED**. The defects on the interface of QD films evolve into the channels of ion migration, degrade the exciton recombination and emissive properties, which would make it poor stability, being the same as the situation observed in analogous film-based solar cells[42,47]. Apart from enhancement in exciton recombination (e.g. PL and EL efficiency), bilateral passivation can enhance the material and device stabilities. We compared the PL attenuation of unilateral passivated, bilateral passivated and pure QD-based films under continuous illumination (365 nm) in ambient air with RH 40%. The PL emission intensity of bilateral-passivated film remained exceed 85% of the original value for 10 h, the unilateral passivated film remained 70%, while the pure QD films lost 60%, as evidenced in Supplementary Fig 12. Furthermore, the operational luminance stability and voltage shift of the devices based on QDs passivated with and without passivation was measured at a constant current density with an initial luminance of about 1000 cd m$^{-2}$ (Fig. 6a, b). The pristine device exhibited a short T$_{50}$ operational lifetime of 1.4 min and a large operational voltage shift. Compared with the control devices, the unilateral passivated devices showed a lower shift in its operational voltage, and exhibited a longer T$_{50}$ of 14 min. Through bilateral passivation, we achieved a great improvement with T$_{50}$ of 30 min, which is 20 times longer than the controlled device. These results demonstrate that bilateral passivation is essential for device stability. And by using the relation L$_0^n$T50 = const. ($n = 1.5$)[48], T50 for controlled and passivated device at 100 cd m$^{-2}$ is predicted to be 47 min and 15.8 h, respectively.

Furthermore, we compared the operational lifetime of controlled and bilateral-passivated device under different initial luminance (Fig. 6c, d). When the initial luminance was 1000 cd m$^{-2}$, the pure QLED exhibited a T50 lifetime of 1.4 min with current density of 10 mA cm$^{-2}$, while the bilateral-passivated device owned a greatly improvement of 30 min with current density of 1.5 mA cm$^{-2}$. Under higher luminance of 5000 cd m$^{-2}$, the lifetime of pure device was 40 s with current density of 84 mA cm$^{-2}$, relatively, the lifetime of passivated one was 7.2 min, where the current density was 10 mA cm$^{-2}$. Moreover, higher brightness at 7000 cd m$^{-2}$ (the highest luminance) and 10000 cd cm$^{-2}$ was also tested for pure QLED and passivated QLED respectively. The unpassivated device showed a quick quenching within 30 s at high current density of 243 mA cm$^{-2}$. The bilateral-passivated one had a T50 of 3.4 min, and the current density under 10,000 cd cm$^{-2}$ luminance was only 30 mA cm$^{-2}$. From the above results, we

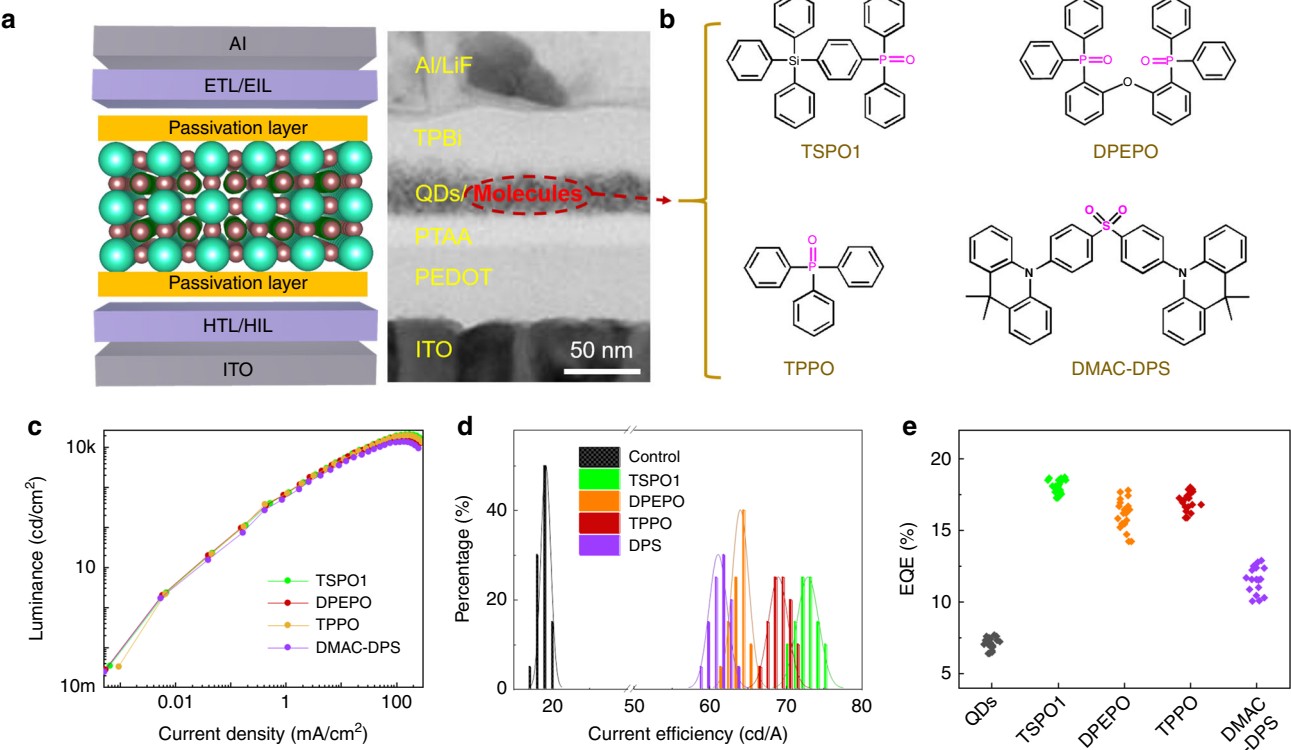

**Fig. 5 EL performance of passivated QLED with various molecules. a** The schematic diagram and sectional TEM image of device structure with bilateral passivation, the scale bar is 50 nm. **b** The molecular structure of the used passivation molecules, TSPO1, DPEPO, TPPO and DMAC-DPS. **c** The luminance vs. current density curves (**d**) current efficiency distribution histogram and (**e**) EQE statistics of devices passivated by TSPO1, DPEPO, TPPO, and DMAC-DPS. Source data are provided as a Source Data file.

| Table 1 Device performance for QLED with different passivation molecules. | | | |
|---|---|---|---|
| Sample | $L_{max}$ (cd cm$^{-2}$) | Current efficiency (cd A$^{-1}$) | EQE (%) |
| Control | 7000 | 20 | 7.7 |
| TSPO1 | 21000 | 75 | 18.7 |
| DPEPO | 20000 | 66 | 17.1 |
| TPPO | 18000 | 71 | 17.8 |
| DMAC-DPS | 13700 | 63 | 12.8 |

could conclude that the higher luminance (higher current density), the faster the device decays for the same device. Compared to unpassivated one, the bilateral-passivated device need less current when reaching the same brightness, which may be also responsible for the better stability, it also illustrates its more effective radiation recombination. In general, bilateral passivation is an effective strategy to enhance the exciton recombination efficiency and increase the material and device stabilities.

**Decreased defects in bilateral-passivated device.** The enhanced exciton recombination of QD films with bilateral passivation (TSPO1 as example) as analyzed in Fig. 7. The defect trap is highly crucial for the exciton recombination. Thus, to accurately evaluate the trap density in these devices, electron-only device with a ITO/TPBi/QDs/TPBi/LiF/Al architecture and hole-only device with a ITO/PEDOT:PSS/PTAA/QDs/TPD/Al architecture were constructed (Fig. 7a, c), the construction details were presented in supporting information. The SCLC method was employed to estimate the trap densities of the QD films[49,50]. The

marked change of the current injection could be used to identify the trap filling process ($I \propto V^n$). As the bias increase, the trap states are gradually filled until reaching the trap-filling limit voltage ($V_{TFL}$). The trap density was calculated by[51]

$$n_{trap} = \frac{2\varepsilon_0 \varepsilon V_{TFL}}{eL^2},  \quad (1)$$

where $\varepsilon$ is the relative dielectric constants of CsPbBr$_3$ ($\approx 22$)[52], $\varepsilon_0$ is the vacuum permittivity, $L$ is the thickness of QD films, and $e$ is the elementary charge. $V_{TFL}$ is obtained by fitting the above $I$–$V$ curves (Fig. 7b, d). The electron trap density for the initial and bilateral-passivated QD films was calculated to be $2.12 \times 10^{18}$ cm$^{-3}$, and $1.05 \times 10^{18}$ cm$^{-3}$, respectively. While, the hole trap density was reduced from $6.7 \times 10^{18}$ cm$^{-3}$ to $3.08 \times 10^{18}$ cm$^{-3}$ after passivation. The passivated QD films presented lower carrier trap density. In addition to SCLC, DLCP method was also applied to test the trap density (Supplementary Fig. 13), which also demonstrated that the trap density of bilateral-passivated device was effectively decreased. From the above results, passivation layer could effectively reduce the defects, which is responsible for reducing non-radiative recombination, resulting in higher radiation efficiency.

The effects of passivation molecules on defect state of QD films are further clarified carrier dynamic under illumination. As schemed in the inset of Fig. 7e, f, before passivation, the defects easily exist at the interface and capture carriers, which would affect the carrier transport channel related to complex trapping/detrapping processes. While, once the defects are passivated, interface would provide a flat channel for carrier transportation. The interfacial defect trapping/detrapping processes can be qualitatively clarified by photo-excited transient current response measurement[53–56]. As shown in Fig. 7e, we could find that photo-generated

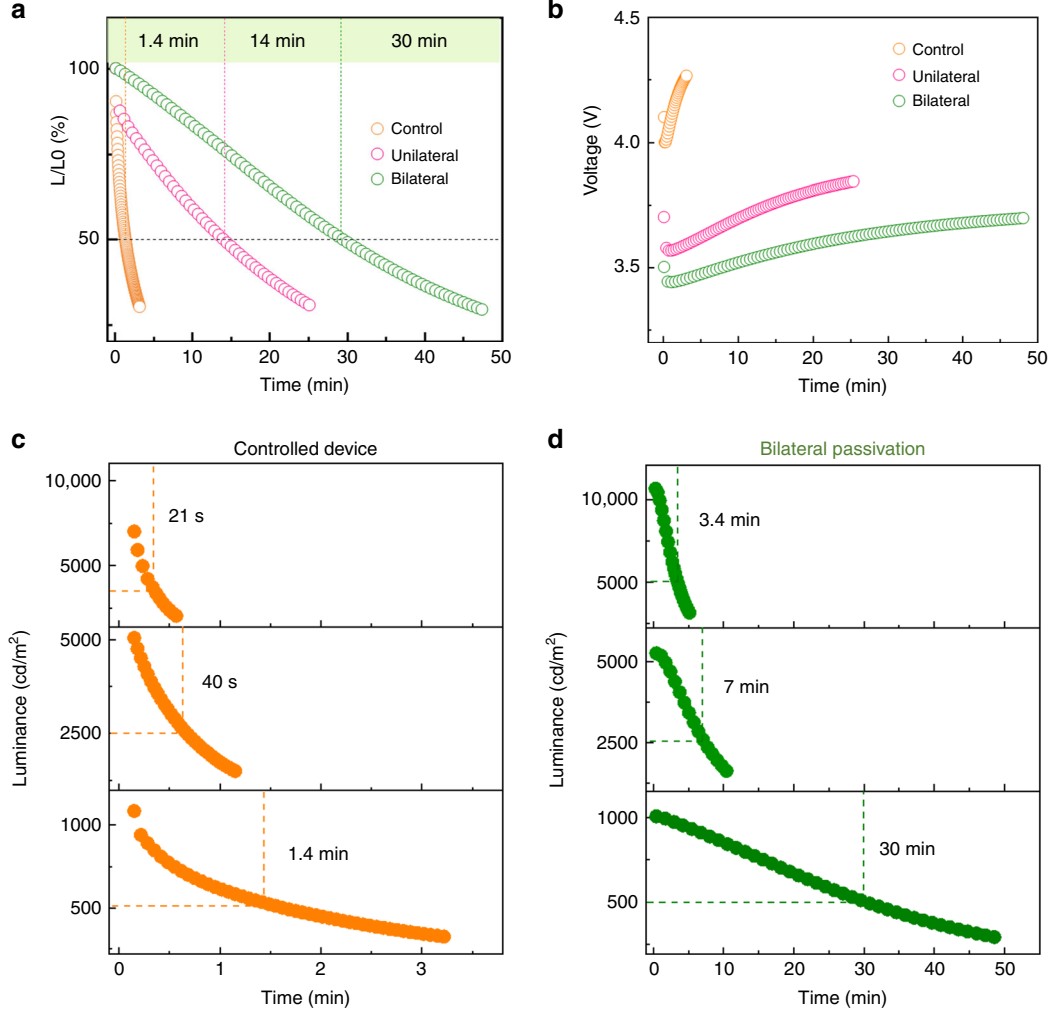

**Fig. 6 Stabilities of QD film and LED device. a** Operational lifetimes of pristine and bilateral-passivated QD films at a initial luminance of about 1000 cd m$^{-2}$. **b** Voltage shifts of perovskite QD LEDs at a constant current density. **c** Operational lifetimes of pristine device tested at different initial luminance, 1000 cd m$^{-2}$, 5000 cd m$^{-2}$, and 7000 cd m$^{-2}$. **d** Operational lifetimes of bilateral-passivated device tested at different initial luminance, 1000 cd m$^{-2}$, 5000 cd m$^{-2}$, and 10000 cd m$^{-2}$. These results demonstrate the bilateral passivation could obviously increase the stability of QD materials and devices. Source data are provided as a Source Data file.

current of the unpassivated QD films showed slower turn-on and turn-off dynamics than that of passivated QD films under the same test condition. The slowly steadying of transient response process (Fig. 7f) in primitive QD films could be ascribed to the time taken for the defect trapping/detrapping processes to reach steady state after under the light switching (turn-on/off). Relatively, the quickly reaching the steady state indicate that the density of defects in passivated QD films is greatly depressed.

## Discussion

In summary, the bilateral passivation strategy demonstrated in this work results in effectively controlling the surface states of QD films. We introduced a layer of organic molecules (e.g. P = O, S = O and C = O) on both top- and bottom-side interface of QD films to reduce the defect density and suppressed the non-radiative recombination. The decreased defects of the QD films are clarified by transient photocurrent measurements, SCLC and DLCP method. The passivated QD films exhibit high exciton recombination features with a PLQY of 79%, and the corresponding LEDs have a high electro-optic conversion efficiency with the EQE of 18.7%. Interestingly, the passivation approach makes the QD materials and LEDs exhibit a higher stability. For

example, a $T_{50}$ operational lifetime of 15.8 h for QLEDs based on QD films passivated by TSPO1 is a factor of 20 longer than the control devices. The proposed bilateral passivation strategy can be widely applied to other types of perovskite materials, and other optoelectronic devices including solar cells, and photodetectors.

## Methods

**Chemicals**. PbBr$_2$ (99.99%), Cs$_2$CO$_3$ (99.9%), oleic acid (OA, AR), oleylamine (OAm, AR) were purchased from Macklin Inc. Hexane, toluene, and ethyl acetate were analytical grade and were purchased from Aladdin Inc. Poly (*bis*(4-phenyl) (2,4,6-trimethylphenyl) amine) (PTAA), and 1,3,5-*Tris*(1-phenyl-1H-benzimida-zol-2-yl)benzene (TPBi) were purchased from Xi'an Polymer Light Technology Corp. (PLT). All reagents were used as received without further purification.

**Synthesis of CsPbBr$_3$ QDs**. 0.2 g PbBr$_2$ were loaded in a 100 mL four-neck flask containing 15 mL of octadecylene (ODE), 3 mL of oleylamine (OAm) and 1.5 mL of oleic acid (OA), after degassed 10 min at 100 °C, maintained at this temperature and continued mixing for 30 min, and then heated to 170 °C in 10 min under Ar flow. 0.55 mL cesium stearate (CsSt)/ODE solution (0.15 M) was injected quickly, 5 s later, the production was rapidly cooled to RT by the ice-water bath. The resultant QDs were precipitated by 40 mL of ethyl acetate and extracted through centrifugation. The collected precipitate was further dispersed in toluene/hexane, add extra ethyl acetate for the second purification, the final precipitate was collected via centrifugation and re-dispersed in *n*-octane/hexane for further use.

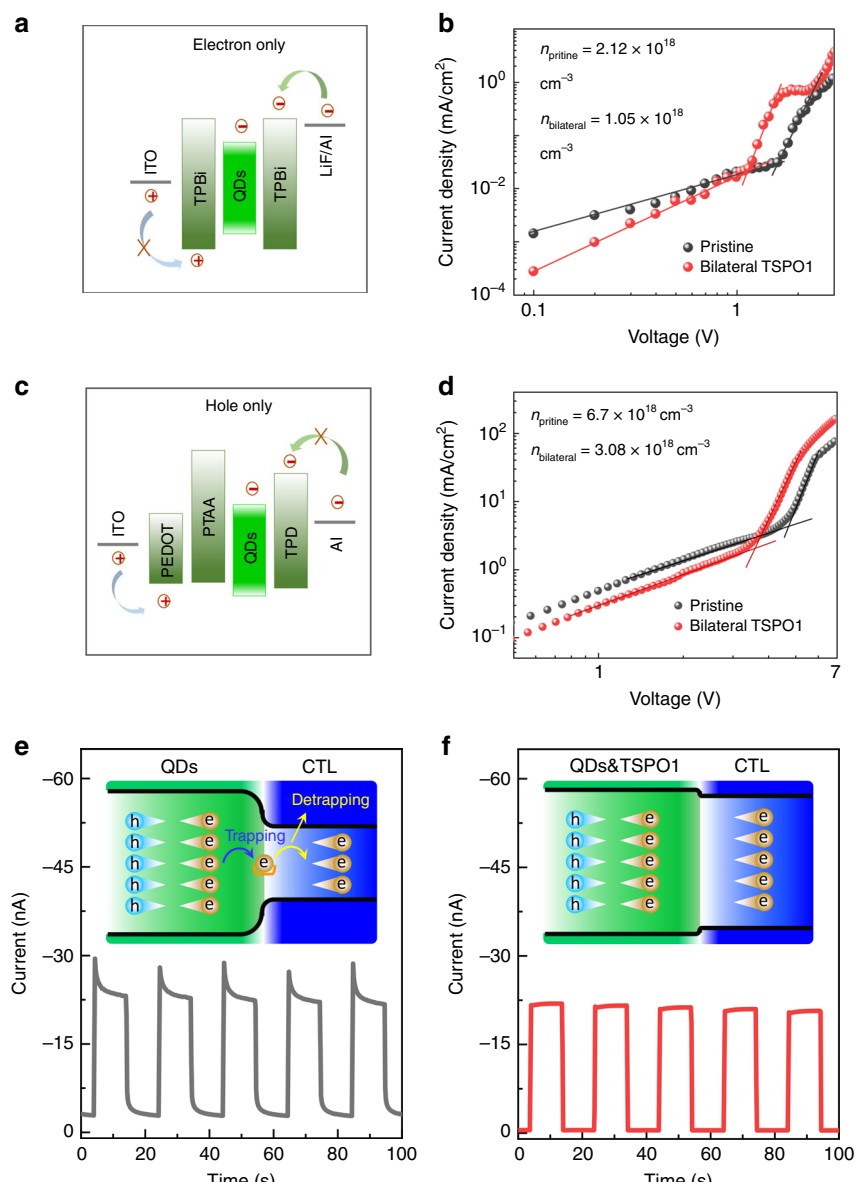

**Fig. 7 Effects of passivation on defect states. a** The structure diagram and corresponding (**b**) current density-voltage curve of electron-only devices based on pristine and passivated QD films. **c** The structure diagram and corresponding (**d**) current density-voltage curve of hole-only devices based on pristine and passivated QD films. Comparison of transient photocurrent of pristine (**e**) and passivated (**f**) QD films. Insets are illustration of carrier-transporting process of pristine and passivated QD films. Before passivation, the defects located on the surface of the QD films or/and between the QD films and carrier-transporting layers, which traps and detraps the carriers, and makes the unsteady current dynamic, especially in the case of light excitation. After passivation, the defects were greatly reduced, and the current is more stable. Source data are provided as a Source Data file.

**Device fabrication**. PEDOT:PSS solutions (Baytron P VPAl 4083, filtered through a 0.22 μm filter) were spin-coated onto the ITO-patterned glass substrates at 4000 r. p.m. for 60 s and baked at 140 °C for 15 min. PTAA (5 mg/mL in chlorobenzene) and CsPbBr$_3$ QDs (20 mg/mL in $n$-octane) were deposited layer-by-layer via spin coating at 2000 r.p.m. for 60 s. Before deposing the next layer, PTAA and QD layers were baked at 120 °C for 15 min and 60 °C for 10 min, respectively. The passivation molecules on the top and bottom of QD films, TPBi (40 nm), and LiF/ Al electrodes (1 nm/100 nm) were deposited through thermal evaporation with a shadow mask under a high vacuum of ~2 × 10$^{-4}$ Pa. The light-emitting area of the device was 4 mm$^2$ as defined by the overlapping area of ITO and Al electrodes.

**Characterization measurements**. The PL spectra of the QD films were obtained by using a Varian Cary Eclipse spectrometer. The PL stability test was performed with a continuous laser under 365 nm excitation at 4 mW cm$^{-2}$ power density. All the samples were tested in air. FTIR measurements were performed using a Shimadzu IR Prestige-21 instrument, with a resolution of 4 cm$^{-1}$. The absolute PLQY was measured using a Horiba Fluorolog system equipped with a single grating and a Quanta-Phil integration sphere coupled to the Fluorolog system. The TRPL decay

lifetimes were acquired via a monochromator/spectrograph (Omni-λ300, Zolix) and an oscilloscope (GDS-3354, GWINSTEK). The ultrafast transient absorption (TA) measurements were performed on a femtosecond (fs) pump-probe system (Helios, Ultrafast System LLC) under ambient conditions. The pump pulses (center wavelength 400 nm, ~20 nJ pulse$^{-1}$ at the sample cell) were delivered by an optical parametric amplifier (TOPAS-800-fs) excited by a Ti:sapphire regenerative amplifier (Legend Elite-1K-HE; 800 nm, 35 fs, 3 mJ pulse$^{-1}$) seeded with a mode-locked Ti:sapphire laser system (Micra-5) and pumped with a Nd:YLF laser (Evolution 30). The time delay between the pump and probe pulses were varied by a motorized optical delay line (maximum ~8 ns). The photo-excited transient response measurement was conducted by using a 442 nm continuous laser controlled by a shutter.

**Device tests**. The EL spectra, $L$–$J$–$V$ characteristics and EQE were collected by using a Keithley 2400 source, a fiber integration sphere, and a PMA-12 spectrometer for light output measurements in glovebox filled with N$_2$ and at room temperature (the measurements equipment is designed by Hamamatsu Photonics Co., Ltd.). T$_{50}$ lifetime is the time over which the device luminance drops to 50% of

the initial value. All of the measurements have been carried out in $N_2$ without encapsulation. Devices were driven by a Keithley 2400 source meter at constant current, and luminance intensity that proportional to photocurrent was measured with a commercial photodiode biased at 0 V.

**First-principles calculation.** First-principles calculation was performed in the framework of density functional theory as implemented in the VASP program. The generalized gradient approximation (GGA) formulated by Perdew, Burke, and Ernzerhof (PBE) was used as the exchange-correlation functional to optimize the structure and simulate density of states (DOS). The electronic wave functions were expended in plane-wave basis sets with a kinetic energy cutoff of 400 eV. The Monkhorst–Pack k-point meshes with a grid spacing of $2\pi \times 0.04$ Å$^{-1}$ or less were used for electronic Brillouin zone integration. The equilibrium structural parameters (including both lattice parameters and internal coordinates) of each involved material were obtained via total energy minimization by using the conjugate-gradient (CG) algorithm, with the force convergence threshold of 0.01 eV Å$^{-1}$. To properly take into account the long-range van der Waals interaction that plays a non-ignorable role in the perovskites involving organic molecules, the DFT-D3 method was adopted. Hybrid functional HSE06 was used to correct DOS results because GGA-PBE usually underestimates conduction bands.

The binding energy $E_b$ is defined as

$$E_b = E_{\text{perovskite+molecule}} - \left( E_{\text{perovskite}} + E_{\text{molecule}} \right),\qquad(2)$$

where $E_{\text{perovskite + molecule}}$, $E_{\text{perovskite}}$, and $E_{\text{molecule}}$ are DFT calculation energies of perovskite anchoring molecule, perovskite, and single P = O molecule, respectively. In addition, $3 \times 3 \times 1$ supercell of perovskite surface model was adopted to screen the interaction between two molecules.

## Data availability
Source data are provided with this paper. Additional data related to this study are available from the corresponding authors on reasonable request.

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

## Acknowledgements

This work was financially supported by NSFC (61725402, 51922049, 61604074), the National Key Research and Development Program of China (2016YFB0401701), the Natural Science Foundation of Jiangsu Province (BK20180020), the Fundamental Research Funds for the Central Universities (30920032102, 30919012107), and PAPD of Jiangsu Higher Education Institutions, the National "ten thousand talents plan" leading talents (No. W03020394), the Six top talent innovation teams of Jiangsu Province (No. TD-XCL-004).

## Author contributions

J.S. and H.Z. conceived the idea, designed the experimental and analyzed the data. L.X. and J.L. synthesized QDs, fabricated devices and collected all data. B.C. provided theoretical calculations and analysis. F.Z. and T.F. were involved in test experiments and data analysis. L.X. and J.L. co-wrote the manuscript. H.Z. directed and supervised the project. All authors contributed to discussions and finalization of the manuscript.

## Competing interests

The authors declare no competing interests.
