## [Peer Review File · Nature Communications]

Reviewers' comments:

Reviewer #1 (Remarks to the Author):

In this manuscript, the authors try to passivate the defect at the both of the top and bottom side perovskite surface, and suppress the non-radiative recombination. Based on this bilateral passivation strategy, QD films passivated by phosphine oxide organic molecule (POOMs) makes the devices achieve a maximum EQE of 18.7% and current efficiency of 75 cd/A, luminance reach 21000 cd/cm². This could be the most efficient of QD perovskite light-emitting diodes. More importantly, due to the substantially binding, the passivation strategy makes the films and LEDs to be of greatly enhanced stability. The results shown in this manuscript are very encouraging for QD perovskite LEDs, although the efficiency is not as high as thin film perovskite LEDs. The paper could be considered for publication if the authors could address following questions:

- 1) QD shows very high PLQY while it is in solution, why the PL will be quenched when forming as solid state? The authors claimed in page 3 lines 69-71: "As is known that a large part of fluorescence is lost when the colloidal QDs transform into the QD solids, this is because massive defects would be introduced inevitably during the film-forming process.", The massive defects could be one reason, another reason could be the aggregation of QDs as I think. The authors should give more explanations.
- 2) P=O bond related molecular have been used for single side passivation of perovskite and enhancing the emission (Nature Photonics, 2018, 12, 355, Nature Communications, 2018, 9, 570). The passivation molecular adopted in this manuscript with the similar key structure. It could be very meaningful if the authors could give a deep mechanism of P=O passivation for perovskite materials.
- 3) The processing details of the passivation molecular such as concentrations, temperature are missing in the experimental parts. And three molecular diphenylphosphine oxide-4-(triphenylsilyl)phenyl (TSPO1), triphenylphosphine oxide (TPPO) and bis(2-(diphenylphosphino)phenyl) ether oxide (DPEPO), why the TSPO1 work the best?
- 4) For the passivation layer deposited on the perovskite, what solvent is used for processing of the molecular such as TSPO1, does this solvent could destroy the perovskite layer?

Reviewer #2 (Remarks to the Author):

In this manuscript, Xu et al. report green-emitting perovskite quantum dot LEDs with bilateral interfacial passivation using phosphine oxide molecules (POOMs), and they achieve a high EQE of 18.7%. This work is interesting, however, the mechanism is not clearly elucidated:

1. It is well-known that P=O endgroup passivates perovskites, including reports such as ACS Energy Letters 2016, 1, 438; Nature Photonics 2018, 12, 355; Nature Communications 2018, 9, 570. The authors use TSPO1, DPEPO and TPPO for passivation, and claim that TSPO1 enables the highest PLQY, but without any rational explanation. What is the difference between various POOMs?
2. It is confusing how POOM can form bonding with the perovskite in different solid layers. The FTIR signal change from 1188 to 1184 cm⁻¹ is quite small and might be an experimental error. The authors should provide stronger evidence to confirm the interaction between POOM and the perovskite.
3. It is confusing why R-P=O is compared with R-SH, R-COOH and R-NH₃ in Figure 1e. How about R-S=O, R-C=O and R-N=O? Can they also passivate QDs?
4. The authors declare that bilateral interfacial passivation decreases the defect density and improves PLQY (Figure 2e). However, they only compare the initial films with unilateral (on the top of QD film)

and bilateral P=O passivation, but without unilateral (on the bottom of QD film). Please provide more comparison between POOM/QD, POOM/QD/POOM and QD/POOM, including but not limited to the PLQY, defect density, crystal structure (TEM), morphology (AFM) and EL performance.

5. TSPO1 and DPEPO are both fluorescent hosts widely used in organic light-emitting diodes and show LUMO energy level of 2.5 eV and HOMO energy level of 6.8 eV. But Figure 4 indicates that they do not affect the energy level of QD films, which is confusing.

6. The authors describe a 15-fold enhancement in the device operational lifetime from 0.8 to 13 hours in the abstract, but not mention the initial luminance. And, what is the lifetime at a higher initial luminance, for example, 1000 cd/m²?

7. In addition, there are some details.

(i) The bond order in Line 129, Page 6 is 2.0, different from the value in Figure 1e.

(ii) The thickness of the QD film in LED should be provided.

(iii) The TTPO should be TPPO in Figure 4h.

(iv) In both the abstract and LED section, the authors show a very high luminance of 21,000 cd/cm², which equals 2.1×10^8 cd/m². But Figure 4c only depicts a luminance lower than 15,000 cd/m², which is confusing.

Reviewer #3 (Remarks to the Author):

This manuscript proposed bilateral interfacial passivation by P=O group to achieve efficient and stable perovskite QLEDs. The novelty is bilateral interfacial passivation, while the P=O passivation method is not new. The EQE of QLEDs here for green LED is still lower than film-based perovskite LEDs, what impressive is the outstanding stability. The authors need to explain why bilateral interfacial passivation can achieve better stability.

1, According to the authors, this work presents 'the most efficient and stable green perovskite QLEDs recorded in this field up to now'; however, considering the results from Lin K, et al, Nature 562, 245-248 (2018) an EQE of >20% is conceivable for green PeLED. First, the authors need to point out the state-of-art perovskite LEDs on any types of perovskite, not just QLEDs. Second, it seems the EQE in this paper of QLEDs is not as high as film-based perovskite LEDs, the authors need to highlight what is the advantage of their green perovskite QLEDs compared to other film-based perovskite LEDs.

2, What is the critical role the passivation plays for the enhancement of device stability should be further analyzed. Can the POOMs prevent the ion migration, or oxygen and water penetration for the prolonged lifetime? Is there any direct evidence for the difference between passivated and unpassivated devices when they reach their T50 lifetime, such as the QDs' structure or surface uncoordinated Pb ions? Related result can help to further understand the decay mechanism of the perovskite QDs and the passivation effect.

3, P=O passivation method by n-triethylphosphine oxide has been reported by Nature Photonics 12, 355-361(2018). The authors should give enough credits to this reference. The main difference here is the bilateral P=O passivation strategy. The P=O passivation strategy is not something new. In order to explain the function of bifacial P=O passivation, the best experimental setup is a comparison of without passivation, passivation at PTAA/QD, passivation at QD/TPBi, and passivation at both PTAA/QD and QD/TPBi for Figure 4 and 5. This can explain whether passivation at both sides are important or just one side is enough.

4, The devices have worked under a bias voltage of ~ 7 V, in this condition, whether the wavelength red-shift appears due to ion-migration or joule heat?

5, The POOMs might promote the injection balance for both carriers (electron and hole injection), or supply a blocking barrier for excess carriers, resulting in the better electrical excitation efficiency. Comparison of the hole-only devices with the electron-only ones might supply a clear explanation in Figure 6.

6, It contains many typos which should be corrected, such as line 21 in page 4, line 7-9 in page 11, and caption of Figure S1, etc. The caption of Figure 1a is not consistent with the image, didn't found any red double-headed arrow in Figure 1a.

Reviewer #4 (Remarks to the Author):

This MS by Xu et al demonstrates a bifacial passivation approach to realize PQDs with very high PLQY and also demonstrates their use as efficient LEDs. The authors perform a lot of measurements and show enhancements in the optical and electronic properties and attribute this to passivation of traps. This is a reasonable study, but I am not sure why this is novel. I do not think this represents the technical innovation required for a MS in Nature comm for the following reasons.

1/ Many groups have demonstrated PQDs pioneered by Kovalenko and co workers, that show near unity PLQY. Why is the starting point of these samples at 40-50%? It is obvious that passivation will work. The important experiment is to try a PDQ with >90% PLQY and then see if this works?

2/ There is a dearth of literature of using surface passivation molecules on low-dimensional hybrid perovskites. Why is this special?

3/ The stability of the LEDs was tested at 120 cd/m². This is where the LED injection current is very low. If the authors claim stability, this should be tested at higher current densities.

Based on these comments, I do not think this MS is suitable for Nature Comm.

Response to Reviewer #1

1. QD shows very high PLQY while it is in solution, why the PL will be quenched when forming as solid state? The authors claimed in page 3 lines 69-71: “As is known that a large part of fluorescence is lost when the colloidal QDs transform into the QD solids, this is because massive defects would be introduced inevitably during the film-forming process.”, The massive defects could be one reason, another reason could be the aggregation of QDs as I think. The authors should give more explanations.

Answer: Thanks very much for the valuable comments. As the reviewer said, both defects and aggregation of QDs are the reasons for the PL quenching. On one hand, the weak binding of surface ligands (such as oleic acid and oleylamine) with perovskite would result in the loss of surface ligands during the solvent evaporation process, which leads to the surface defects. These defects can become the non-radiative centers to trap electrons. The weak binding was confirmed by DFT simulation, which was also reported before (ACS Appl. Mater. Interfaces 2015, 7, 25007–25013). On the other hand, the aggregations of QDs can cause the loss of quantum confinement, leading to the PL quenching as well. But I think defects are the main reason in this case. The aggregation phenomenon is more obvious in colloid QDs when QDs are over-purified (ACS Appl. Mater. Interfaces 2015, 7, 25007–25013; Adv. Mater. 2017, 29, 1603885). In that situation, the QD film would be non-uniform and broken (Figure R1), and QDs would integrate into small clusters on the film. While, the colloidal QDs we use here uniformly monodisperse, the QDs would not cluster together, and the QD film is uniform (Figure R2a). Therefore, the effect of the particle aggregation is not primary.

[Redacted]

Figure R1 The colloidal QDs and QD films under different times of purification
(*Advanced Materials*, 2017, 29(5): 1603885.)

Figure R2 Photographs of (a) Colloidal QDs and (b) QD film 你 this work, (c) TEM
image of corresponding QD

2. P=O bond related molecular have been used for single side passivation of perovskite and enhancing the emission (Nature Photonics, 2018, 12, 355, Nature Communications, 2018, 9, 570). The passivation molecular adopted in this manuscript with the similar key structure. It could be very meaningful if the authors could give a deep mechanism of P=O passivation for perovskite materials.

Answer: This advice is very helpful for our work. We have carefully studied these two documents, indeed, P=O bond related molecules have been reported for single side passivation of perovskite. However, previous work use these molecules based on solution method, that is, molecules are dissolved in a solvent and then spin-coated on the perovskite film. As is known that QD based film is sensitive to organic solvents, which would destroy the film. In our work, we deposit the passivation molecules by thermal evaporation, which effectively avoid the effects of solvents. Meanwhile, few work paid attention to bilateral passivation, here we proposed a bilateral passivation strategy with P=O, but not limited to P=O.

As for the mechanism of P=O passivation for perovskite materials, we will elaborate from the following aspects.

First, from the molecular coordination aspect, ligands with P=O (such as TOPO) were widely used to passivate QDs (*ACS Omega* 2019, 4, 9150–9159) and QD films (*Nat. Commun.* 2018, 9, No. 570), where the electronegative oxygen bonded with uncoordinated Pb^{2+} , thus decreasing surface defects of QD or QD film. Here, the interaction between P=O with perovskite is speculatively feasible through theoretical calculation. The forming energy between Pb and O from TSPO1 was -1.1 eV calculated by density functional theory, which showed that the interaction between surface Pb and O=P could be easily formed. Meanwhile, the FTIR and XPS tests also demonstrate that this interaction does exist experimentally as shown in Figure R3. FTIR peak (Figure R3a) located at 1188 cm^{-1} was observed for TSPO1, corresponding to P=O bond stretching vibrations. The peak shifted to 1184 cm^{-1} , this shift indicated the interaction of P=O with QDs. Figure 3b showed the Pb 4f peak shifts towards higher binding energy of about 0.2 eV for TSPO1-passivated QD films in reference to unpassivated one, which unveiled the chemical bonding between P=O group and Pb atom. The electronegative O⁻ of TSPO1 would coordinate with Pb^{2+} .

Figure R3 (a)The FTIR and (b)XPS spectra of pure QD film and TSPO1-passivated QD film

Secondly, from the improvement on the optical electrical properties, on account of the coordination between P=O with uncoordinated Pb²⁺ on QD surface, the defects could be effectively passivated, thus decreased the non-radiative recombination, leading to enhanced PL and EL efficiency.

The decreased trap density in PL was verified through ultrafast exciton dynamics as shown in Figure R4. Figure R4a and b revealed the global analysis of the pristine and TSPO1-passivated QD film, where τ_1 was intraband hot-exciton relaxation, τ_2 was exciton trapping to the band-edge trap states, and τ_3 exciton recombination. The former two processes were accelerated for the passivated sample relative to the unpassivated one, indicating that the TSPO1 promoted the state coupling involved in the relaxation processes. Thus the TSPO1 passivated QD films exhibited a slower kinetic recombination delay (Figure R4c), which reflected a lower density of surface defect trap states in the films.

Figure R4 (a) and (b) Decay-associated spectra for pristine and TSPO1-passivated QD films. (c) Comparison of transient TA spectra with an excitation fluence of $5 \mu\text{J cm}^{-2}$. Inset: Schematic illustration of the photoinduced relaxation processes involved in the QD films.

The decreased defects were evidenced by electron-only and hole-only device. We designed the electron-only and hole-only device as shown in Figure R5a and R5c. The space charge-limited-current method (*J. Am. Chem. Soc.* 2018, 140, 1358-1364) was used to evaluate the electron trap density and hole trap density. Eventually, the electron trap density and hole trap density of the pristine device were $2.12 \times 10^{18} \text{ cm}^{-3}$, $6.7 \times 10^{18} \text{ cm}^{-3}$, and that of passivated device were $1.05 \times 10^{18} \text{ cm}^{-3}$, $3.08 \times 10^{18} \text{ cm}^{-3}$ (Figure R4b and R4d). The trap density was indeed decreased.

Eventually, the improved radiative recombination was reflected on enhanced PLQY and EL efficiency. The PLQY of QD film was increased from 43% to 79%, and EQE was enhanced from 7.7% to 18.7%.

Correspondingly, the description and data have been added into the revised

manuscript and supporting materials.

Figure R5 The structure diagram of (a)electron-only and (c)hole-only devices. Current density-voltage curve of (b)electron-only and (d)hole-only devices based on pristine and passivated QD films.

3. The processing details of the passivation molecular such as concentrations, temperature are missing in the experimental parts. And three molecular diphenylphosphine oxide-4-(triphenylsilyl)phenyl (TSPO1), triphenylphosphine oxide (TPPO) and bis(2-(diphenylphosphino)phenyl) ether oxide (DPEPO) , why the TSPO1 work the best?

Answer: I am very grateful for your reminding and that is our omission. But in fact, the passivation molecular here was deposited through thermal evaporation, not spin coating. So there are no parameters such as concentration, temperature, etc. And the processing details were presented in the experimental section, “The passivation molecules on the top and bottom of QD films, TPBi (40 nm), and LiF/Al electrodes (1 nm/100 nm) were deposited using a thermal evaporation system through a shadow mask under a high vacuum of $\sim 2 \times 10^{-4}$ Pa.”

As for the difference of the three passivation molecules, although P=O is the key factor for the effective passivation, the effect of different molecules is discrepant due to their different structure, conductivity, and other intrinsic factors. We designed the electrode-only device of ITO/POOMs/Au (Figure R6a) to compare their electrical transport properties, the I-V curves of TSPO1, DPEPO and TPPO were presented in Figure R6b. It could be seen that TSPO1 exhibit the best conductivity versus DPEPO and TPPO. In sum, all three organic molecules have a great passivation effect for the P=O, while the relatively good conductivity of TSPO1 makes it more beneficial for the charge transfer, thus works the best.

Correspondingly, the description and data have been added into the revised manuscript and supporting materials.

Figure R6 (a) The electrode-only device structure, (b) I-V curves of TSPO1, DPEPO and TPPO

4. For the passivation layer deposited on the perovskite, what solvent is used for processing of the molecular such as TSPO1, does this solvent could destroy the perovskite layer?

Answer: I am sorry that we did not describe it clearly and misled you. Actually, the organic passivation molecules were deposited on the substrate through thermal evaporation, not by solution process. Profiting from this thermal evaporation method, we have avoided the influence of solvents on QD film. In order to avoid misunderstanding, we have modified the experimental section and emphasized the deposition method in the text, which are shown as following.

Page 5 line 106, “we present a bilateral passivation strategy to reduce the interfacial defects by **evaporating a layer of organic molecules** between QD films and carrier transport layer (CTL).”

Response to Reviewer #2

In this manuscript, Xu et al. report green-emitting perovskite quantum dot LEDs with bilateral interfacial passivation using phosphine oxide molecules (POOMs), and they achieve a high EQE of 18.7%. This work is interesting, however, the mechanism is not clearly elucidated:

1. It is well-known that P=O endgroup passivates perovskites, including reports such as ACS Energy Letters 2016, 1, 438; Nature Photonics 2018, 12, 355; Nature Communications 2018, 9, 570. The authors use TSPO1, DPEPO and TPPO for passivation, and claim that TSPO1 enables the highest PLQY, but without any rational explanation. What is the difference between various POOMs?

Answer: Thanks for your very useful suggestion, these work are very helpful for our work. In fact, surface ligands with P=O endgroup have been widely applied in perovskites, which also proves that P=O bond is indeed an effectual passivation group for perovskites. In these reported cases, the P=O endgroup was used in interface passivation on the top surface of perovskite film, no one has paid attention to the case of bilateral passivation. In this work, we are trying to propose a bilateral passivation strategy for perovskite-based device, not limited to P=O endgroup, other ligands with passivation effect will be also applicable.

As for the difference of the three molecules, on the whole, all of them have a positive effect on the QD film and device, the slight difference is due to their different molecular structure and electrical properties. The molecular structure of them is shown in Figure R7. The better PLQY of TSPO1 passivated film maybe because of the larger branched chains, which could better prevent external environment from damaging the film. It has been reported that complex branched ligands exhibit stronger steric hindrance, which can prevent QDs from external environment and lead to better PL properties and stabilities (Angew. Chem, 2016, 128, 1–6; Mater. Res. Express, 2016, 3, 095903; J. Phys. Chem. Lett, 2015, 6, 5027).

Figure R7 The molecular structure of TSPO1, DPEPO and TPPO

Furthermore, the better EL performance of TSPO1 is due to the better conductivity. We designed the electrode-only device of ITO/Molecules/Au (Figure R8a) to compare their electrical transport properties, the I-V curves of TSPO1, DPEPO and TPPO were presented in Figure R8b. It could be seen that TSPO1 exhibit the best conductivity versus DPEPO and TPPO. In sum, all three organic molecules have a great passivation effect for the P=O, while the relatively good conductivity of TSPO1 makes it more beneficial for the charge transfer, thus works the best.

Correspondingly, the description and data have been added into the revised manuscript and supporting materials.

Figure R8 (a) The electrode-only device structure, (b) I-V curves of TSPO1, DPEPO and TPPO

2. It is confusing how POOM can form bonding with the perovskite in different solid layers. The FTIR signal change from 1188 to 1184 cm⁻¹ is quite small and might be an experimental error. The authors should provide stronger evidence to

confirm the interaction between POOM and the perovskite.

Answer: The ligands with P=O (such as TOPO) were widely used to passivate QDs (*ACS Omega* 2019, 4, 9150–9159) and QD films (*Nat. Commun.* 2018, 9, No. 570), where the electronegative oxygen bonded with uncoordinated Pb^{2+} , thus decreasing surface defects of QD or QD film. Here, the interaction between P=O with perovskite is speculatively feasible through theoretical calculation, meanwhile, the FTIR and XPS tests also demonstrate that this interaction does exist experimentally.

First, we established the theoretical model of PbBr_2 -rich CsPbBr_3 surface, to calculate the bonding ability of perovskite with POOM. The forming energy between Pb and O from TSPO1 was -1.1 eV calculated by density functional theory, which showed that the interaction between surface Pb and O=P could be easily formed.

As for the reviewer's doubt about the FTIR, we have retested the FTIR spectra data in order to eliminate the experimental errors, as shown below (Figure R9). The FTIR signal change from 1208 to 1200 cm^{-1} , the characteristic peak shifts slightly due to the changed test environment though, the relative positions of these two peaks do not change much. So I think the FTIR signal change would not be caused by the experimental error, but the evidence of the interaction between POOM and perovskite. The shift of P=O bond peak means that the formation of coordination bonds between Pb=O and QDs.

Furthermore, the XPS (Figure R10) can also demonstrate Pb 4f peak shifts towards higher binding energy of about 0.2 eV for TSPO1-passivated QD films in reference to unpassivated one, which unveiled the chemical bonding between P=O group and Pb atom, which is due to the coordination between electronegative O⁻ and uncoordinated Pb^{2+} , leading to higher binding energy for Pb 4f. I hope our answer dispels your doubts.

Correspondingly, the description and data have been added into the revised manuscript and supporting materials.

Figure R9 Rested FTIR spectra of TSPO1, pure QD and TSPO1 passivated QD film

Figure R10 XPS of TSPO1, pure QD and TSPO1 passivated QD film

3. It is confusing why R-P=O is compared with R-SH, R-COOH and R-NH₃ in Figure 1e. How about R-S=O, R-C=O and R-N=O? Can they also passivate QDs?

Answer: Your question is very valuable. The several groups we compared are the common ligands used in QDs, which were not selected randomly. For example, oleic acid (R-COOH) and oleylamine (R-NH₃) are widely used in perovskite QDs. In our case, R-COOH and R-NH₃ are the surface ligands wrapped on the perovskite QDs, the R-COOH and R-NH₃ exhibit weak interaction with perovskite, which leads to the

easy loss of surface ligands. In contrast, R-P=O has stronger bond with perovskite, which is responsible for the better stability.

The bilateral passivation strategy is universal, not just P=O based ligands, I think other suitable materials are also applicable here, such as the reviewer mentioned, R-S=O, R-C=O and R-N=O. To verify the universal strategy, we found a organic molecules with S=O endgroup, DMAC-DPS (the molecular structure is shown in Figure R11), which

Figure R11 The molecular structure of DMAC-DPS

was also applied in the bilateral-passivated QLED. The device data were presented in Figure R12. It could be seen that DMAC-DPS passivation had a good optimization effect on the nude QD-based device. The DMAC-PDS passivated LED shows similar phenomenon as the POOM passivated device. From the results, not only P=O bond, other functional group can also passivate QD film. And we have proposed a universal bilateral passivation strategy.

Correspondingly, the description and data have been added into the revised manuscript and supporting materials.

Figure R12 (a) current density, (b)EQE and (c) luminance of DMAC-DPS passivated and nude QD LED

4. The authors declare that bilateral interfacial passivation decreases the defect density and improves PLQY (Figure 2e). However, they only compare the initial films with unilateral (on the top of QD film) and bilateral P=O passivation, but without unilateral (on the bottom of QD film). Please provide more comparison between POOM/QD, POOM/QD/POOM and QD/POOM, including but not limited to the PLQY, defect density, crystal structure (TEM), morphology (AFM) and EL performance.

Answer: Thanks for your helpful advises, we have supplemented relevant data and readjusted the article based on your suggestions. The major modifications are as follows:

1) We have supplemented the emission photographs, PL spectra and PLQY of QD, TSPO1/QD, QD/TSPO1, TSPO1/QD/TSPO1 film, as shown in Figure R13. The PL intensity and PLQY can be improved through unilateral passivation. Compared to bottom-side passivation, the top-side passivation on QD film showed brighter emission and higher PLQY. That was because the upper surface was exposed, and the top-side passivation would be more productive. Nevertheless, the bilateral passivation could achieve the optimal results.

Correspondingly, the description and data have been added into the revised manuscript and supporting materials. The additional data were presented in revised Figure 2, and the specific modifications were reflected on page 7 and page 8 of the revised manuscript.

Figure R13 The (a) photograph, (b) PL spectra and (c) PLQY of pure, with TSP01 on the bottom side, on the top side, on both sides of QD film

2) We have also increased the EL performance of unilateral QD/TSP01 passivation, based LEDs, shown in Figure R14. It could be seen that the addition of passivation layer could decrease the electrical properties of the device due to the relative poor carrier mobility. However, the luminance and EL efficiency were effectively increased on account of higher exciton recombination efficiency. The unilateral passivation can improve the EL performance to some degree, which demonstrated the passivation was feasible. Compared to the bottom-side passivation, the top-side passivation showed better improving effect, this was because TSP01 was a carriers transport material that was partial electronic (*Organic Electronics* 2015, 23, 138–143). Thus the passivation on the electron-side performed better. Although the unilateral passivation had made a certain majorization, it was not as effective as the bilateral passivation. After bilateral passivation, the EQE was increased from 7.7% to 18.7%, the luminance was increased from 7000 cd/cm² to 21000 cd/cm², and current efficiency was increased from 23 cd/A to 75 cd/A.

Correspondingly, the description and data have been added into the revised manuscript and supporting materials. Relative data were added in revised Figure 4, the modifications were presented in page 9 and page 10 of the new manuscript edition.

Figure R14 The EL performance of CsPbBr₃ QLED under unilateral passivation with different thickness and bilateral passivation

3) In addition, we also retested the EL stabilities of the QD, unilateral-passivated (on the top side) and bilateral-passivated device, which were shown in Figure R15. It could be seen that unilateral passivation can also improve the stabilities of the QLED, which could also testify that the interface passivation was a good way to improve

stability. But obviously, the improvement of bilateral passivation is better. Similarly, the PL stability was also supplemented as shown in Figure R 16, which showed the same trend as EL stability.

Correspondingly, the description and data have been added into the revised manuscript and supporting materials. The supplements were reflected in revised Figure 6 and Figure S10.

Figure R15 The EL stabilities of pure CsPbBr₃ QLED, and QLED under unilateral passivation, bilateral passivation respectively

Figure R16 The PL stabilities of pure CsPbBr₃ QD film, and QD film under unilateral passivation, bilateral passivation respectively

From above results, interface passivation is actually an effective way to improve the performance of perovskite QD based device, however the bilateral passivation can bring this effect to extreme.

5. TSPO1 and DPEPO are both fluorescent hosts widely used in organic light-emitting diodes and show LUMO energy level of 2.5 eV and HOMO energy level of 6.8 eV. But Figure 4 indicates that they do not affect the energy level of QD films, which is confusing.

Answer: Actually, TSPO1, DPEEPO and TPPO are all fluorescent hosts, where the TSPO1 and DPEEPO show LUMO energy level of 2.5 eV and HOMO energy level of 6.8 eV, TPPO shows LUMO energy level of 7.1 eV and HOMO energy level of 1.1 eV. Since the passivation layer on the surface of perovskite film is very thin (2 nm), it should affect the energy level of the interface, but would not affect that of the perovskite subject. We tested the energy level of pure QD film and passivated QD film through UPS and UV-vis absorption, as shown in Figure R17 and R18. The tested VB and CB values are listed in Table R1. It could be seen that the optical band gap of the films have not changed before and after passivation, the VB and CB of the passivated films move upward very slightly. In brief, these organic molecules have a slight effect on the energy level of QD films, but the effect is not very large, so we ignore it.

Figure R17 UPS of nude QD, TSPO1 passivated QD, DPEPO passivated QD and TPPO passivated QD

Figure R18 UV-vis absorption spectra of nude QD, TSPO1 passivated QD, DPEPO passivated QD and TPPO passivated QD

Table R1 The VB and CB value of naked QD film and passivated QD film

Sample	VB	CB
Naked QD film	5.41	3.07
TSPO1/QD film	5.43	3.09
DPEPO/QD film	5.43	3.09
TPPO/QD film	5.44	3.1

6. The authors describe a 15-fold enhancement in the device operational lifetime from 0.8 to 13 hours in the abstract, but not mention the initial luminance. And, what is the lifetime at a higher initial luminance, for example, 1000 cd/m²?

Answer: We appreciate your suggestions very much. Actually, we tested the lifetime with an initial luminance of about 120 cd m⁻². As the reviewer mentioned, we only tested the stability under a very low initial luminance, to better expound the stability issue, we retested the EL stabilities of nude QLED and passivated QLED under higher initial luminance of 1000 cd/m², as shown in Figure R19. The controlled sample exhibited a T50 lifetime of 1.4 minutes, while the passivated device achieved an improvement to 14 minutes for unilateral passivation, and 30 minutes for bilateral

passivation. The predicted lifetime of nude device was 47 minutes and the bilateral-passivated one was 15.8 h, which were approximately close to our previous tested result.

Figure R19 The operational lifetime of controlled, unilateral-passivated and bilateral-passivated device with initial luminance of 1000 cd/m^2

Furthermore, we also tested the operational lifetime of nude device and bilateral-passivated device at even higher luminance of 5000 cd/m^2 and 10000 cd/m^2 (the nude QLED was tested at the highest luminance, 7000 cd/m^2), as shown in Figure R20. Under 5000 cd/m^2 , the lifetime of pure device was 40 s, relatively, the lifetime of passivated one was 7.2 minutes. Higher brightness at 7000 cd/m^2 (the highest luminance) and 10000 cd/cm^2 for pure QLED and passivated QLED respectively, the unpassivated device showed a quick quenching within 30 s, while the bilateral-passivated one had a T50 of 3.4 minutes. It could be seen that bilateral passivation could greatly improve the operational stabilities of perovskite QLED.

Correspondingly, the description and data have been added into the revised manuscript and supporting materials. The specific modifications were presented in revised Figure R6, page 11 and page 12 of the updated manuscript.

Figure R20 The operational lifetime of (a) controlled device and (b) bilateral-passivated device at different initial luminance.

7. In addition, there are some details.

(i) The bond order in Line 129, Page 6 is 2.0, different from the value in Figure 1e.

(ii) The thickness of the QD film in LED should be provided.

(iii) The TTPO should be TPPO in Figure 4h.

(iv) In both the abstract and LED section, the authors show a very high luminance of 21,000 cd/cm², which equals 2.1×10⁸ cd/m². But Figure 4c only depicts a luminance lower than 15,000 cd/m², which is confusing.

Answer: I am regretful for our carelessness. We have double-checked our manuscript carefully, and seriously revised these detail errors. The modification list is as follows.

1) Line 129, “bond order is 2.0” changed to “bond order is 0.2”.

2) The thickness of the QD film is about 23 nm as shown in the cross-section diagram

(Figure R21). The cross-section diagram was added in the revised Figure 5.

Figure R21 The cross-section diagram of perovskite QLED device.

3) The typo in Figure 4h has been corrected.

4) As for the luminance of 21000 cd/cm^2 , which is the highest luminance under bilayer passivation. We have modified Figure 4, which exhibits the EL performance of unilateral (on the top of QD film), unilateral (on the bottom of QD film), bilateral passivated device. .

Response to Reviewer #3

This manuscript proposed bilateral interfacial passivation by P=O group to achieve efficient and stable perovskite QLEDs. The novelty is bilateral interfacial passivation, while the P=O passivation method is not new. The EQE of QLEDs here for green LED is still lower than film-based perovskite LEDs, what impressive is the outstanding stability. The authors need to explain why bilateral interfacial passivation can achieve better stability.

1. According to the authors, this work presents 'the most efficient and stable green perovskite QLEDs recorded in this field up to now'; however, considering the results from Lin K, et al, Nature 562, 245-248 (2018) an EQE of >20% is conceivable for green PeLED. First, the authors need to point out the state-of-art perovskite LEDs on any types of perovskite, not just QLEDs. Second, it seems the EQE in this paper of QLEDs is not as high as film-based perovskite LEDs, the authors need to highlight what is the advantage of their green perovskite QLEDs compared to other film-based perovskite LEDs.

Answer: Your suggestions are very helpful to our work, this is indeed our mistake in presentation. We have corrected our presentation as “Here, we propose a bilateral passivation strategy, inserting ultra-thin organic molecule layers into the both upside and downside interfaces, to sufficiently and substantially passivate the surface defects, and hence drastically enhance the efficiency and stability of perovskite QLEDs.”

Meanwhile, according to the reviewer’s advice, we have readjusted the introduction part in the revised manuscript. We have added the current progress in the field of perovskite LEDs, not just the QLEDs.

As for the advantage of the perovskite QLEDs. Indeed, film-based LEDs have achieved a very high efficiency (>20%). But QLEDs also have made great breakthrough in recent years, the red perovskite QLEDs have also exceeded 20% (*Nature Photonics*, 2018, 12, 681-687), the green QLEDs are slightly behind, only 16.8% (*Adv. Mater.* 2018, 1805409), but the I think breakthrough of 20% is already on the way. Meanwhile, colloidal QDs have better flexibility, film-formation process is

less affected by the environment, the QD filming process does not even require heat treatment. And colloidal QDs have good process compatibility, which are suitable for varieties of solution film-forming process, such as roll to roll, ink-jet printing, and film quality can be guaranteed once the QDs are synthesized. QD-based LED have greater potential in industrial production. In general, film-based LED and QD-based LED are two important branches of perovskite in LED field, both of which possess their own advantages. The advancement of both has crucial significance for the further development of perovskite in display field.

Correspondingly, the description and data have been added into the revised manuscript and supporting materials.

2. What is the critical role the passivation plays for the enhancement of device stability should be further analyzed. Can the POOMs prevent the ion migration, or oxygen and water penetration for the prolonged lifetime? Is there any direct evidence for the difference between passivated and unpassivated devices when they reach their T50 lifetime, such as the QDs' structure or surface uncoordinated Pb ions? Related result can help to further understand the decay mechanism of the perovskite QDs and the passivation effect.

Answer: Thanks for your valuable advice. As for the enhancement of device stability, I think there are the following factors.

First, the P=O bond of POOMs could passivate the surface defects of QD film, which could provide channels for ion migration. Surface and interface defects were reported to be important ion migration channels (*Acc. Chem. Res.* 2016, 49, 286–293), which is an important factor in accelerating the degradation of perovskite-based device. Here, the defects were effectively decreased by bilateral passivation, as shown in Figure R22.

Figure R22 (a) The electron-only device structure and corresponding (b) I-V characteristics. (a) The hole-only device structure and corresponding (b) I-V characteristics.

We designed the electro-only and hole-only devices to estimate the electron trap density and hole trap density by the space-charge-limited-current (SCLC) technique (*J. Appl. Phys.*, 1962, 33, 1733-1737). The electron and hole trap density of pristine device was $2.12 \times 10^{18} \text{ cm}^{-3}$ and $6.7 \times 10^{18} \text{ cm}^{-3}$, while that of the bilateral-passivated device were decreased to $1.05 \times 10^{18} \text{ cm}^{-3}$ and $3.08 \times 10^{18} \text{ cm}^{-3}$ respectively. It could be seen that defects were actually reduced, which was beneficial to decrease the ion migration channels and enhance the stability.

Secondly, the passivation molecules could also form a barrier layer to prevent the erosion by water and oxygen. During device operation, trace amount of water and oxygen can damage the device (*Adv. Mater.* 2018, 1704587) can damage the device, leading to poor stability. After passivation, the water and oxygen stability has been

greatly improved, that can be observed from Figure R 23. Figure R16 represents the PL attenuation of QD film under the atmosphere with RH 40%. It can be seen that the ability resisting wet and oxygen of QD film is greatly improved after bilateral passivation.

Figure R23 The PL attenuation of QD film under the atmosphere with RH 40%.

The above are the main reasons for the improved device stability. As for the direct evidence for the difference between passivated and unpassivated devices when they reach their T50 lifetime, it's too difficult to test. Because QD layer is sandwiched between the HTL and ETL, it is difficult to characterize the surface conditions of QD layer under working states. In addition, we had tried to observe the cross section, but the thickness of QD film is too thin, subtle changes are barely visible. So unfortunately, we cannot give direct evidence for the difference between passivated and unpassivated devices when they reach their T50 lifetime so far. However, I believe that the bilateral passivation strategy we have proposed is a good exploration to improve stability. And as the reviewer proposed, the decay mechanism and promotion mechanism of the perovskite QD based LED are also worth further excavation. We will explore further in subsequent work.

Correspondingly, the description and data have been added into the revised manuscript and supporting materials.

3. P=O passivation method by n-trioctylphosphine oxide has been reported by Nature Photonics 12, 355–361(2018). The authors should give enough credits to this reference. The main difference here is the bilateral P=O passivation strategy. The P=O passivation strategy is not something new. In order to explain the function of bifacial P=O passivation, the best experimental setup is a comparison of without passivation, passivation at PTAA/QD, passivation at QD/TPBi, and passivation at both PTAA/QD and QD/TPBi for Figure 4 and 5. This can explain whether passivation at both sides are important or just one side is enough.

Answer: Thanks for your sincere suggestions, these references are very helpful for us. Indeed, P=O passivation is not new, some work have applied P=O based ligands (TOPO) in perovskite to improve radiative recombination or device efficiency (*Nature Photonics 2018, 12(6): 355-361; Nat Commun 2018, 9(1): 570*)^{1, 2} as the reviewer mentioned. These also testified that ligands with P=O group were effective passivators for perovskite based device. However, their passivation strategies only focused on the top interface of QD film, few mentioned bilateral passivation. Here, we are trying to propose a universal bilateral passivation strategy, P=O is one of the effective means. Not only P=O, other effectual passivators are also applicable. Anyway, this suggestion is very helpful for improving our work. As requested, we have supplemented the relevant EL performance and stability data, specific descriptions are as follows.

We have also increased the unilateral QD/TSP01 passivation, based LEDs, shown in Figure R24. It could be seen that the addition of passivation layer could decrease the electrical properties of the device due to the relative poor carrier mobility. However, the luminance and EL efficiency were effectively increased on account of higher exciton recombination efficiency. The unilateral passivation can improve the EL performance to some degree, which demonstrated the passivation was feasible. Compared to the bottom-side passivation, the top-side passivation showed better improving effect, this was because TSP01 was a carriers transport material that was partial electronic (*Organic Electronics 2015, 23, 138–143*). Thus the passivation on the electron-side performed better. Although the unilateral passivation had made a certain majorization, it was not as effective as the bilateral passivation. After bilateral passivation, the EQE was increased from 7.7% to 18.7%, the luminance was increased from 7000 cd/cm² to 21000 cd/cm², and current efficiency was increased from 23

cd/A to 75 cd/A.

Relative data were added in revised Figure 4, the modifications were presented in page 9 and page 10 of the new manuscript edition.

As for the stability, we retested the EL stabilities of nude QLED, unilateral-passivated (on the top side) QLED and bilateral-passivated QLED under higher initial luminance of 1000 cd/m², as shown in Figure R25a and b. The unpassivated device exhibited a short T50 lifetime of 1.4 minutes, and the passivated device achieved an improvement to 14 minutes for unilateral passivation, while the lifetime could be increased to 30 minutes under bilateral passivation. It could be seen that although unilateral passivation can improve the EL stability, but it is far worse than bilateral passivation.

Therefore, bilateral passivation is very necessary. The supplemental data were presented in revised Figure 6.

From the above, the interface passivation on the QD film is very effective in improving device performance and stability. However, passivation on both interfaces of the QD film performs better than only passivated one side.

Correspondingly, the description and data have been added into the revised manuscript and supporting materials.

Figure R 24 The EL performance of CsPbBr₃ QLED under unilateral passivation with different thickness and bilateral passivation

Figure R25 The operational lifetime of pristine device and passivated device.

4. The devices have worked under a bias voltage of ~7 V, in this condition, whether the wavelength red-shift appears due to ion-migration or joule heat?

Answer: The EL spectra of nude QD-based and passivated QD-based devices under different voltage are presented in Figure R26. It could be seen that no red-shift appears with EL peaks at 517 nm when voltage increases. However, this does not mean no ion migration occurs during the operation. The ion-migration or joule heat may not hamper the EL, but they could cause the device deterioration (*Adv. Mater.* 2018, 1704587). And the spectra shift may be more common in mixed halogen perovskites (such as $\text{CsPb}(\text{Cl},\text{Br})_3$ and $\text{CsPb}(\text{Cl},\text{Br})_3$), where the changes of the halide composition will affect the spectral position. Here is the single component halogen, CsPbBr_3 , the spectra shift is not obvious.

Correspondingly, the description and data have been added into the revised manuscript and supporting materials.

Figure R26 EL spectra of nude QD based (a) and POOM passivated QD based (b) LED under different voltage

5. The POOMs might promote the injection balance for both carriers (electron and hole injection), or supply a blocking barrier for excess carriers, resulting in the better electrical excitation efficiency. Comparison of the hole-only devices with the electron-only ones might supply a clear explanation in Figure 6.

Answer: Thanks for your valuable advice. The electron-only device was actually used

to further verify the decreased trap density after effective passivation, which was were estimated by the space-charge-limited-current (SCLC) technique (*J. Appl. Phys.*, 1962, 33, 1733-1737). Through this method, the electron trap density of pristine device was $2.12 \times 10^{18} \text{ cm}^{-3}$, and that of bilateral-passivated device was $1.05 \times 10^{18} \text{ cm}^{-3}$. It could be seen that the electron trap density was reduced after passivation. However, as the reviewer said, the hole trap density should also be provided to reveal the defects. Therefore, we designed the hole-only device with structure, ITO/PEDOT:PSS/PTAA/QDs/TPD/Al, as shown in Figure R27a. The I-V characteristics of hole-only devices were shown in Figure R27b, the hole trap density of pristine and passivated device was $6.7 \times 10^{18} \text{ cm}^{-3}$ and $3.08 \times 10^{18} \text{ cm}^{-3}$ respectively. It could be seen that bilateral passivation could also effectively passivate hole traps. The reduction of defects is responsible for reducing non-radiative recombination, resulting in higher radiation efficiency.

Figure R27 (a) The electron-only device structure and corresponding (b) I-V characteristics. (a) The hole-only device structure and corresponding (b) I-V characteristics.

Figure R28 Current density–voltage curves of electron-only and hole-only devices under bilateral passivation

Moreover, as the reviewer requested, we compared the hole-only device with the electron-only device in Figure R28. The POOMs can promote both electrons and holes, but the promotion of electrons is better than holes, which is also the reason that the passivation on the top side performs better than that on the bottom side.

Correspondingly, the description and data have been added into the revised manuscript and supporting materials. Relevant modifications are presented in revised Figure 6.

6, It contains many typos which should be corrected, such as line 21 in page 4, line 7-9 in page 11, and caption of Figure S1, etc. The caption of Figure 1a is not consistent with the image, didn't found any red double-headed arrow in Figure 1a.

Answer: Sorry for our negligence. We have corrected these typos in the revised manuscript, and we have checked the text repeatedly to avoid such errors again. We list several modifications following:

- 1) The caption of Figure S1 has been corrected.
- 2) Figure 1a, we have added red double-headed arrow in revised Figure 1a.

Response to Reviewer #4

This MS by Xu et al demonstrates a bifacial passivation approach to realize PQDs with very high PLQY and also demonstrates their use a efficient LEDs. The authors perform a lot of measurements and show enhancements in the optical and electronic properties and attribute this to passivation of traps. This is a reasonable study, but I am not sure why this is novel. I do not think this represents the technical innovation required for a MS in Nature comm. for the following reasons.

1. Many groups have demonstrated PQDs pioneered by Kovalenko and coworkers, that show near unity PLQY. Why is the starting point of these samples at 40-50% ? It is obvious that passivation will work. The important experiment is to try a PDQ with >90% PLQY and then see if this works?

Answer: Thanks for your suggestion, but our statements might cause you some misunderstandings. The PLQY we talk about in Figure 2e was the PLQY of the solid QD film, not colloidal QDs, the PLQY of the colloidal in our work was 85±3%. Indeed, for colloidal PQDs, near unity PLQY has been reported by a lot of work. However, when colloidal QDs transform into solid film, a large part of PLQY will be lost, which is a common problem in the field. For example, Kido group also reported the same situation in CsPb(Br/I)₃ QDs (*Nature Photonics*, 2018, 12(11): 681), the QDs in a toluene dispersion exhibited PLQY of 80%, while the QD film decreased to 26%. Akkerman and his workers (*Nature Energy*, 2016, 2(2): 1-7) also pointed out the problem of low PLQY of perovskite film, and they proposed a novel surface treatment method that improved the PLQY of perovskite QD film into 35%. Kovalenko group also (*JACS*, 2016, 138(43): 14202-14205) pointed out the dramatically decrease of PLQY from solution state to film state. Chen (*Adv. Optical Mater.* 2018, 1800007) reported a surface-passivated method, where the colloidal QDs exhibited a high PLQY of 85% but the PLQY of QD film was only 23.6%.

It could be seen that the decreased PLQY of solid QD film compared to colloidal QDs was a common problem. Therefore, we proposed the bilateral passivation strategy, which could effectively increase the PLQY of QD film. Figure R29 exhibits the passivation diagram of pure QD film, and QD film with TSPO1 passivated on bottom side, top side and both sides of the QD layer. It could be seen that the brightness was greatly enhanced under bilateral passivation, which was due to the interface defects

could be effectively passivated by TSPO1.

Figure R29 QD films without passivation and with TSPO1 on the bottom side, on the top side, on both sides of QD film under UV light.

The PLQY of unpassivated and passivated QD films were tested and shown in Figure R30. From the result, unilateral passivation on top side can improve the PLQY from 43% to around 70%, while the bilateral passivation can further improve it to 79%. So in general, the original PLQY in this work was at an average level, and the value was improved to a higher level under bilateral passivation. However, in order not to cause similar misunderstandings, we have modified the Figure 2e, where we add a schematic vignette of film excitation in revised edition.

Correspondingly, the description and data have been added into the revised manuscript and supporting materials.

Figure R30 PLQY of QD film under different passivation state

2. There is a dearth of literature of using surface passivation molecules on low-dimensional hybrid perovskites. Why is this special?

Answer: The reviewer has proposed a very good point. I think the feasibility of the passivation strategy is also related to the structural characteristics of the materials. For the low-dimensional hybrid perovskites, they generally consist of 2D perovskite structure and long-chain organic amines on both sides, as shown in Figure R31 (*Nature Communications*, 2018, 9, 570). The long-chain organic amines are the key to ensuring the low-dimensional structure. And the long-chain organic amine not only participates in the structure of low-dimensional perovskites, but also acts as a surface ligand. So there is no need to use surface passivation molecules on low-dimensional hybrid perovskites.

[Redacted]

Figure R31 The structure of low-dimensional hybrid perovskites

However, for QDs, additional surface ligands are important factors in maintaining their colloidal stability and passivating the surface defects. In general, surface ligands do not participate in structural composition of QDs, but only act as surface passivators. Nonetheless, ligands adsorbed on the perovskite QD surface is out of a dynamic equilibrium process, which means that the surface ligands are easily lost. Meanwhile, perovskite is ionic crystal, the surface ions is easy to be lost and cause surface mismatches, resulting in defects. These features make perovskite QD surface a crucial issue. Therefore, surface passivation strategy is an important means to improve the properties of QD-based devices. Due to the issue, the surface and interface problems of QDs and QD based device have attracted spread attention, but most of the work focused on the top-side interface of the film, the bilateral passivation strategy has not been proposed. Here, we found that not only top-side of the film, both interfaces of perovskite film need to be taken seriously.

Correspondingly, the description and data have been added into the revised manuscript and supporting materials.

3. The stability of the LEDs was tested at 120 cd/m². This is where the LED injection current is very low. If the authors claim stability, this should be tested at higher current densities.

Answer: Thanks for your valuable advice. Actually, the previous stability test at 120 cd/m² was too low. To better illustrate the stability, we retested the stabilities of the QLED devices at higher luminance, and we also modified the manuscript. First, we tested the operational lifetime and the voltage shift of controlled, unilateral-passivated and bilateral-passivated devices at a initial luminance of 1000 cd/m², the lifetime curves were presented in Figure R32. The controlled device showed current density of 10 mA/cm², and exhibited a fast degradation with a short T50 of 1.4 minutes. The T50 of unilateral passivated device were increased to 14 minutes with the current density of 4.4 mA/cm². While, under bilateral passivation, the lifetime was improved to 30 minutes, where the current density was 1.5 mA/cm². It could be seen that the bilateral passivation plays a crucial function on the stabilities.

Figure R32 The operational lifetime of controlled, unilateral-passivated and bilateral-passivated device with initial luminance of 1000 cd/m²

Secondly, we also tested the operational lifetime of nude device and bilateral-passivated device under different brightness, as shown in Figure R33. When

the initial luminance was 1000 cd/m^2 , the pure QLED exhibited a T50 lifetime of 1.4 minutes (current density was 10 mA/cm^2), while the bilateral passivated was 30 minutes (current density was 1.5 mA/cm^2). Under higher luminance of 5000 cd/m^2 , the lifetime of pure device was 40 s (current density was 84 mA/cm^2), relatively, the passivated one was 7.2 minutes (current density was 10 mA/cm^2). Moreover, higher brightness at 7000 cd/m^2 (the highest luminance) and 10000 cd/cm^2 was also tested for pure QLED and passivated QLED respectively. The unpassivated device showed a quick quenching within 30 s at high current density of 243 mA/cm^2 . The bilateral-passivated one had a T50 of 3.4 minutes, and the current density under 10000 cd/cm^2 luminance was only 30 mA/cm^2 .

For the same device, higher current may accelerate the attenuation of device. From above results, we found that the bilateral-passivated device need less current when reaching the same brightness, which may be also responsible for the better stability, it also illustrates its more effective radiation recombination. The deeper discussion on the stability issue will be explored in another work, which is under way.

In summary, the bilateral passivation provides a promising way to improve the stability of perovskite based devices.

Correspondingly, the description and data have been added into the revised manuscript and supporting materials.

Figure R33 The operational lifetime of (a) controlled device and (b) bilateral-passivated device at different initial luminance.

Reviewers' comments:

Reviewer #1 (Remarks to the Author):

The authors have tried their best to address the issues mentioned by the Reviewers, the paper can be accepted for publication now.

Reviewer #2 (Remarks to the Author):

The authors have addressed major comments. However these concerns remain in the revised work:

1. The authors use vacuum-evaporated TSPO1, DPEPO and TPPO to passivate the perovskite QD layer, but spin-cast TPPO has been recently reported to passivate reduced-dimensional perovskites (Nature Communications, 2020, 11, 170), so I am not sure whether the present submission is novel enough.

2. The authors now use DMAC-DPS (S=O) for comparison. I wonder whether there is an extra emission peak from DMAC-DPS in EL spectra of the corresponding LEDs? How about other C=O or N=O molecules? Can they also passivate QDs? Please provide more evidence of the passivation mechanism.

3. The authors now provide comparison between POOM/QD, POOM/QD/POOM and QD/POOM, but one concern is that the solvent to spin-coat QD, i.e. n-octane, may dissolve POOM and destroy the device structure. How did the authors address this problem?

4. The authors should provide details of fabricating electron-only devices, and explain how they prepare QD layer using n-octane on TPBi without destroying the lower film.

Reviewer #3 (Remarks to the Author):

The authors responded referee's comments in detail with more interesting results. They also have made a lot of revision in the revised version. I think the article is sufficient for its acceptance in the journal after solving some remaining problems:

1. Figure R16, R23 and Figure S10 are the same figure, but the caption and description on Figure R16 and R23 in the rebuttal file is not consistent with the related Figure S10 in Supporting Information and Revised Manuscript.

Figure R16: The PL stabilities of pure CsPbBr₃ QD film, and QD film under unilateral passivation, bilateral passivation respectively

Figure R23: The PL attenuation of QD film under the atmosphere with RH 40%.

Figure S10: The PL stability of pristine and bilateral passivated QD films excited by 365 nm.

2. could the trap density be quantified by other widely used methods other than the space-charge-limited-current (SCLC) technique? such as Drive-level capacitance profiling (DLCP)?

Reviewer #4 (Remarks to the Author):

This paper by Xu et al describes the role of bifacial passivation on perovskite QD leds. Authors have

responded to many questions raised by the reviewers, which is highly appreciated. However, after going over all the questions raised by the referees and looking at some of the results, I do not still believe that this paper adds a substantially to the large literature on perovskite based LEDs. The authors report a high QE approaching 20% but the stability is of the order of minutes, even after passivation. The stability of perovskite LEDs is the biggest open question and this has not been addressed. Also, this has been measured at 1000 Cd while the peak is 21000 Cd. This is 1/20 th of the efficiency.

There are also measurements on measuring the times resolved PL with and without passivation and these hardly change in their lifetime. In fact, if the authors fit the curve, they will find that the lifetime is identical. Also, the defect density measured is $2 \times 10^{18}/\text{cm}^3$, which is high. The authors claim a reduction in defect density from 2 to $1 \times 10^{18}/\text{cm}^3$. This is within error. This suggests that the passivation does not seem to affect the charge recombination within the film.

I do not see what is the new science that would move the field of perovskite based LEDs forward and therefore do not believe that this paper should be published in Nature Communications.

Reviewer #1

Agree to acceptance, No question.

Reviewer #2

1. The authors use vacuum-evaporated TSPO1, DPEPO and TPPO to passivate the perovskite QD layer, but spin-cast TPPO has been recently reported to passivate reduced-dimensional perovskites (Nature Communications, 2020, 11, 170), so I am not sure whether the present submission is novel enough.

Answer: Your advice is very useful. We have carefully read this article, which has applied phosphine oxides in reduced-dimensional perovskite for better stabilities. Actually, the phosphine oxygen ligands used to passivation perovskite is not new, which has been widely used in perovskite-based films (*Nature photonics*, 2018, 12, 355–361), LEDs (*Nature communications*, 2018, 9, 1-8) and solar cells. However, in this work we have proposed a novel interface passivation strategy, the bilateral passivation strategy, with passivation molecules on both top-side and bottom-side interface of perovskite film for perovskite-based device. The passivation molecules were not limited to P=O ligands, as we have presented in the manuscript (Figure 5), S=O could also be applied in this strategy. In addition, we also applied C=O (benzophenone) and N=O (nitrosobenzene) in the bilateral-passivated device. Our experimental results showed that N=O would destroy the QD film, while C=O could also improve the device performances. The EL performances of device with and without C=O passivation are presented in Figure R1, it can be seen that C=O can also enhance the efficiency and brightness of the LED.

Since interface passivation has always been an effective means to improve the performance of planar structure device, but most of them focus on only one side of the interface. Generally, perovskite material is in the sandwich position in the optoelectronic device, which owns the top-side and bottom-side interface, both interfaces. Interface problems have always been big problems in thin film based devices, both the top- and bottom-side interfaces are worthy of attention. Therefore, it is necessary to passivate the top-side and bottom-side interface of the perovskite film, whether it is to passivate defects, inhibit ion migration, or improve the device from the perspective of energy level adjustment. Here we propose a bilateral passivation

strategy, which provides a novel idea for improving the performance of perovskite-based optoelectronic devices.

According to the recommendations, we have modified the expression and highlighted in red “Thus, interface passivation of QD film is particularly important. Here, we propose a bilateral passivation strategy through passivating both top- and bottom-side interface of perovskite QD film with organic molecules, which has drastically enhanced the efficiency and stability of perovskite QLEDs. Various molecules were applied in this method, and the comparison of unilateral and bilateral passivation were conducted to verify the necessity of passivation on both interfaces of QD film” in abstract. Supplementary data about C=O and N=O are added to Figure S10 and S11, corresponding descriptions “a series of organic molecules with various functional groups, including DPEPO, TPPO (~P=O), DMAC-DPS (~S=O), nitrosobenzene (~N=O) and benzophenone (~C=O), were also applied in the bilateral passivated device structure” and “The schematic structures of these organic molecules are listed in Figure 5b and S10. In addition to N=O would seriously damage the perovskite QDs and QD film, other molecules could improve the QLED performances to varying degrees (Figure 5c-d and S11), of which P=O and S=O performed better” are added to page 11.

Figure R1 EL performances of pure CsPbBr₃ QLED and bilateral-passivated QLED

with benzophenone

2. The authors now use DMAC-DPS (S=O) for comparison. I wonder whether there is an extra emission peak from DMAC-DPS in EL spectra of the corresponding LEDs? How about other C=O or N=O molecules? Can they also passivate QDs? Please provide more evidence of the passivation mechanism.

Answer: Thank you for your valuable advice. The EL spectra of the DMAC-DPS passivated LED is presented in Figure R2. It could be seen that no extra emission peak is observed, and the peak position is at around 517 nm, which exhibits no shift compared to pure QLED.

Figure R2 The EL spectra of the DMAC-DPS passivated LED

As for the C=O and N=O molecules, we chose nitrosobenzene (with N=O) and benzophenone (with C=O) as typical sample to passivate perovskite, the molecular structures of which were presented in Figure R3. We found that nitrosobenzene (with N=O) could not be used to passivate QDs, while benzophenone (with C=O) could passivate QDs.

The passivated QDs with N=O were presented in Figure R4, it could be seen that N=O severely damaged QDs, which almost lose the fluorescence. It may damage the perovskite structure, so nitrosobenzene (with N=O) can not be used to passivate the perovskite QLEDs.

Figure R3 The molecular structure of nitrosobenzene and benzophenone

Figure R4 The fluorescent photographs of QDs and passivated QDs with nitrosobenzene

Nevertheless, the passivated QDs/film with benzophenone (C=O) exhibited brighter emission compared to original QDs, as shown in Figure R5. Furthermore, we applied C=O in bilateral-passivated CsPbBr₃ QLED. The EL performances were also improved as presented in Figure R6, the current density of passivated device exhibited lower electrical properties for the poor carrier mobility (Figure R6a). While, the luminance was enhanced for higher exciton recombination efficiency (Figure R6b). And the EQE and current efficiency of passivated device was also improved compared to controlled one on account of effective passivation, as shown in Figure R6c and 6d. It could be seen that benzophenone can available passivate the perovskite QD film and improve the radiation recombination efficiency, thus leading to enhanced QLED performances. But the improving effect is not as great as P=O.

Figure R5 The fluorescent photographs of QDs/film and passivated QDs film with benzophenone

Figure R6 EL performances of pure CsPbBr₃ QLED and bilateral-passivated QLED with benzophenone

As we can see from above results, the organic molecule with C=O (benzophenone) we choose has a positive effect on perovskite QDs and QLED device, however, the nitrosobenzene (with N=O) damages the perovskite QDs. In this work, we focus on proposing a bilateral passivation strategy for perovskite QLEDs, and various molecules, P=O, S=O, C=O and N=O have been applied in this system. From the experimental results, except for N=O, the first three all have positive effects on

QLED. Therefore, as long as the appropriate ligands are selected, they can be applicable in this bilateral passivation strategy. In this work, P=O based ligands were regarded as the typical sample, which interaction mechanism with perovskite were analyzed in detail. However, studying the interaction mechanism between each kind of ligands and perovskite surface is not the focus of this work, I think it is worth digging into a lot of research.

Supplementary data about C=O and N=O are added as Figure S10 and S11 in supporting material, and the relevant expressions are added to page 11 in revised manuscript.

3. The authors now provide comparison between POOM/QD, POOM/QD/POOM and QD/POOM, but one concern is that the solvent to spin-coat QD, i.e. n-octane, may dissolve POOM and destroy the device structure. How did the authors address this problem?

Answer: Your question is very valuable, which is also the challenge that is often encountered in the preparation of multi-layered devices by solution method. So the choice of solvent is very important. In this work, QDs were dispersed in n-octane, however, these POOMs, TSPO1, DPEPO and TPPO, can hardly dissolve in n-octane, thus, spin-coating QDs would not dissolve the POOM. In addition, the POOM layer was deposited through thermal evaporation, which possessed better compactness and uniformity than spin-coated film. Therefore, spin-coating QD layer will not greatly damage the underlying POOM. It should be noted that the spin-coating process should be conducted as soon as colloidal QDs come into contact with the underlying film, avoiding staying too long.

Furthermore, from the obtained device results, it can also be seen that spin-coating QDs has not affected the enhancement effect of the underlying POOM on the QLEDs. Figure R7 presents the EL performance of original CsPbBr₃ QLED and with different thickness of POOM on the bottom side of QD film. It can be seen that the passivated devices with 1 nm or 2 nm POOM on the bottom side of QD film can effectively improve the EL performance of original device, which also demonstrates that spin-coating QDs has little effect on underlying POOM layer.

Figure R7 The EL performance of original CsPbBr₃ QLED and with different thickness of POOM on the bottom side of QD film

4. The authors should provide details of fabricating electron-only devices, and explain how they prepare QD layer using n-octane on TPBi without destroying the lower film.

Answer: Thanks for your suggestions. The details of fabricating electron-only devices, which consists of ITO/TPBi/QDs/TPBi/LiF/Al, are added in supporting information: “After the ITO-coated glass substrate was cleaned and sonicated, TPBi (40 nm) was deposited on the substrate using a thermal evaporation system through a shadow mask under a high vacuum of $\sim 2 \times 10^{-4}$ Pa, Then CsPbBr₃ QDs were deposited by spin coating at 2000 r.p.m. for 60 s, QD layers were baked at 60 °C for 10 min. Finally, TPBi (40 nm) and LiF/Al electrodes (1 nm/100 nm) were deposited through thermal evaporation. The annotations in the text are described as “the construction details were presented in supporting information” in page 13

As for the doubt about whether the the spin-coated quantum dot process would damage the underlying TPBi, there were some tips. TPBi was not soluble in octane, and evaporated TPBi was not so easily washed away. However, we found that colloidal QDs stayed on the TPBi film too long would leave traces, leading to poor test results. Therefore, when QD layer was deposited, the colloidal QDs (here the

solvent was octane) should be spin-coated as soon as possible after dripping on the ITO/TPBi substrate to minimize the contact time. In addition, we also found that the solvent resistance of TPBi film was different under different evaporator, which may be due to the different surface wettability caused by different roughness of formed films. So these problems could be solved by controlling the spin-coating process and the evaporation environment.

Reviewer #3

1. Figure R16, R23 and Figure S10 are the same figure, but the caption and description on Figure R16 and R23 in the rebuttal file is not consistent with the related Figure S10 in Supporting Information and Revised Manuscript.

Figure R16: The PL stabilities of pure CsPbBr₃ QD film, and QD film under unilateral passivation, bilateral passivation respectively

Figure R23: The PL attenuation of QD film under the atmosphere with RH 40%.

Figure S10: The PL stability of pristine and bilateral passivated QD films excited by 365 nm.

Answer: Thank you for your careful checking, we are very sorry for our negligence. Actually, the PL stability was tested under continuous illumination (365 nm) in ambient air with RH 40%. We have corrected the description in the manuscript and supporting information, the modification is as follows.

Page 12 Line 289 in manuscript: “We compared...QD based films under continuous illumination in ambient air” is changed to “We compared...QD based films under continuous illumination (365 nm) in ambient air with RH 40%”.

Figure S10 in supporting information: “The PL stability of pristine and bilateral passivated QD films excited by 365 nm” is changed to “The PL stability of pristine unilateral passivated and bilateral passivated QD films under continuous illumination (365 nm) in ambient air with RH 40%”

2. could the trap density be quantified by other widely used methods other than the space-charge-limited-current (SCLC) technique? such as Drive-level capacitance profiling (DLCP)?

Answer: Thank you for your valuable suggestion. Actually, SCLC method is widely used to test defect states in QLEDs (*J. Am. Chem. Soc.* 140, 1358-1364 (2018), *Adv. Mater.*, 1803019 (2018).). We have tried our best to do the DLCP measurement with the QLED device structure. Capacitance with respect to amplitude was obtained to correct the capacitance value at higher orders, following the equation (1)

$$C = C_0 + C_1 dV + C_2 (dV)^2 + \dots \quad (1)$$

The density of states (DOS) for the devices with variation in frequency was calculated

using the following equation (2):

$$N = -\frac{C_0^3}{2q\varepsilon A^2 C_1} \quad (2)$$

C_0 and C_1 are obtained by quadratic polynomial fitting of equation 1, q is the elementary charge, ε is the relative dielectric constants of CsPbBr₃, A is the junction area. The DOS of both pure device and bilateral-passivated device from DLCP are presented in Figure R7. As reported, at high modulation frequency, only the intrinsic free carriers contribute to DOS, whereas at low modulation frequency, both intrinsic free carriers and trap states contribute to DOS (*Adv. Mater.* 2017, 1605756; *Adv. Mater.* 2015, 27, 4481–4486). It could be seen that the trap density of passivated device was greatly decreased by 1.5 times compared to unpassivated one, which was consistent with the SCLC test result.

The supplementary data are added to Figure S13, the computing method are presented below Figure S13, and relevant descriptions have been presented in page 14 “In addition to SCLC, DLCP method was also applied to test the trap density (Figure S13), which also demonstrated that the trap density of bilateral passivated device was effectively decreased.”.

Figure R8 Density of states of pristine perovskite device and bilateral-passivated device with TSPO1 from DLCP measurement

Reviewer #4:

The authors report a high EQE approaching 20% but the stability is of the order of minutes, even after passivation. The stability of perovskite LEDs is the biggest open question and this has not been addressed. Also, this has been measured at 1000 Cd while the peak is 21000 Cd. This is 1/20 th of the efficiency.

There are also measurements on measuring the times resolved PL with and without passivation and these hardly change in their lifetime. In fact, if the authors fit the curve, they will find that the lifetime is identical. Also, the defect density measured is $2 \times 10^{18}/\text{cm}^3$, which is high. The authors claim a reduction in defect density from 2 to $1 \times 10^{18}/\text{cm}^3$. This is within error. This suggests that the passivation does not seem to affect the charge recombination within the film

Answer: We appreciate the reviewer's comments. **With regard to the stability**, at present, the operation stability of perovskite QD based LED is indeed a problem, which dilemma it faces is much more severe than that of film based device. The reported EL lifetimes of the QD based LED are presented in Table R1. **It could be seen that most of the operational lifetimes were within 10 h, some even were of the order of seconds. In this work, the L_{50} of the passivated QLED was improved to 15.8 h at initial luminance of 100 cd m^{-2} , although the stability was not very satisfactory, but it has made some progress compared to existing work.** There are not much researches on the stability of perovskite QLEDs so far, and there is still a long way to go. This work just has only taken a small step on the stability issue, while this work is not focused on stability. Our work on stability has made a little breakthrough and will be reported in another work later.

Table R1 The reported operational lifetime of perovskite QLEDs

Refs	Emission layer	Lifetime	Test conditions
Phys. Status Solidi RRL 2020, 2000083	CsPbBr ₃ NCs	$L_{50} = 955 \text{ s}$	$L_0 = 520 \text{ cd m}^{-2}$
Nature Photonics , 2018, 12(11): 681-687	CsPb(Br/I) ₃ QDs	$L_{50} = 180 \text{ min}$	$L_0 = 100 \text{ cd m}^{-2}$

Nano Lett. 2017, 17, 313	CsPbBr ₃ NCs	L ₈₀ = 9 h	At 10 V
J. Mater. Chem. C 2017, 5, 4565	CsPbBr ₃ NCs	L ₇₀ = 1.75 h	--
Adv. Mater. 2017, 29, 1606405	FA _{0.8} Cs _{0.2} PbBr ₃ NCs	L ₅₀ = 80 s	--
Nano Lett. 2016, 16, 1415–1420	CsPbBr ₃ NCs	L ₅₀ = 10 min	At 5 v

As for the query about times resolved PL, the reviewer think the lifetime is identical after fitting. Actually, the fitting result of PL lifetime was presented in supporting information (Figure S4). As shown in Figure R9, the lifetime is obviously different, the passivated QD film shows average lifetime of 13.9 ns, while the unpassivated one shows only 6.7 ns, the lifetime is greatly improved.

Figure R9 The PL decay curves of primal and TSPO1-passivated CsPbBr₃ QD films.

Regarding the reviewer's doubt about the defect density measurement, the results show that the defect density was reduced by half, which were not test error. Previous report by Wei Huang and Feng Gao (*Nature Photonics* 2019, 13, 418–424) also only showed a decrease of the defect density from 7.8×10^{15} to 5.9×10^{15} after treatment, which is less than half. Furthermore, the defect density was confirmed by DLCP method, the DOS vs. frequency was shown in Figure R10, the defect density

was indeed decreased after passivation.

Figure R10 Density of states of pristine perovskite device and bilateral-passivated device with TSPO1 from DLCP measurement

In addition, the reviewer concludes that passivation does not affect the charge recombination within the film. However, whether from the perspective of the defect state tests, exciton dynamics analysis, or from the performances of QD film and QLED devices, this conclusion is untenable. **Firstly**, as we have discussed above, the defect density was indeed decreased. **Secondly**, both the ultrafast exciton dynamics analysis and time-resolved PL demonstrated prolonged exciton lifetime after passivation, as presented in Figure 3. **Thirdly**, PLQY of QD film was enhanced from 43% to 79%, the EQE of QLED was improved from 7.7% to 18.7%, while the current efficiency was increased from 20 to 75 cd/A and luminance increased from 8000 to 21000 cd/cm². These data all reflect the enhanced recombination within the film.

In conclusion, we have proposed a novel bilateral passivation strategy for perovskite QLEDs, the sufficient passivation on both interfaces can effectively reduce defect density, improve radiative recombination efficiency and enhance device performances.

REVIEWERS' COMMENTS:

Reviewer #2 (Remarks to the Author):

The authors addressed my concerns. The work is suited for publication in Nature Comms.

Reviewer #1

No question.

Reviewer #2 (Remarks to the Author):

The authors addressed my concerns. The work is suited for publication in Nature Comms.

Agree to acceptance, No question.

Reviewer #3

No question.

Reviewer #4

No question.